# RT-Sketch: Goal-Conditioned Imitation Learning from Hand-Drawn Sketches

## Abstract

Natural language and images are commonly used as goal representations in goal-conditioned imitation learning (IL). However, natural language can be ambiguous and images can be over-specified. In this work, we study hand-drawn sketches as a modality for goal specification. Sketches are easy for users to provide on the fly like language, but similar to images they can also help a downstream policy to be spatially-aware and even go beyond images to disambiguate task-relevant from task-irrelevant objects. We present RT-Sketch, a goal-conditioned policy for manipulation that takes a hand-drawn sketch of the desired scene as input, and outputs actions. We train RT-Sketch on a dataset of paired trajectories and corresponding synthetically generated goal sketches. We evaluate this approach on six manipulation skills involving tabletop object rearrangements on an articulated countertop. Experimentally we find that RT-Sketch is able to perform on a similar level to image or language-conditioned agents in straightforward settings, while achieving greater robustness when language goals are ambiguous or visual distractors are present. Additionally, we show that RT-Sketch has the capacity to interpret and act upon sketches with varied levels of specificity, ranging from minimal line drawings to detailed, colored drawings. For supplementary material and videos, please refer to our website.[1]

## 1 Introduction

Robots operating alongside humans in households, workplaces, or industrial environments have an immense potential for assistance and autonomy, but careful consideration is needed of what goal representations are easiest *for humans* to convey to robots, and *for robots* to interpret and act upon.

Instruction-following robots attempt to address this problem using the intuitive interface of natural language commands as inputs to language-conditioned imitation learning policies (Brohan et al., 2023b;a; Karamcheti et al., 2023; Lynch & Sermanet, 2020; Lynch et al., 2023). For instance, imagine asking a household robot to set the dinner table. A language description such as *"put the utensils, the napkin, and the plate on the table"* is under-specified or ambiguous. It is unclear how exactly the utensils should be positioned relative to the plate or the napkin, or whether their distances to each other matter or not. To achieve this higher level of precision, a user may need to give lengthier descriptions such as *"put the fork 2cm to the right of the plate, and 5cm to the leftmost edge of the table."*, or even online corrections (*"no, you moved too far to the right, move back a bit!"*) (Cui et al., 2023; Lynch et al., 2023). While language is an intuitive way to specify goals, its qualitative nature and ambiguities can make it both inconvenient for humans to provide without lengthy instructions or corrections, and for robot policies to interpret for downstream precise manipulation.

On the other hand, using goal images to specify objectives and training goal-conditioned imitation learning policies either paired with or without language instructions has shown to be quite successful in recent years (Jiang et al., 2022; Jang et al., 2022). In these settings, an image of the scene in its desired final state could fully specify the intended goal. However, this has its own shortcomings: access to a goal image is a strong prior assumption, and a pre-recorded goal image can be tied to a particular environment, making it difficult to reuse for generalization.

---

[1] http://rt-sketch-anon.github.io

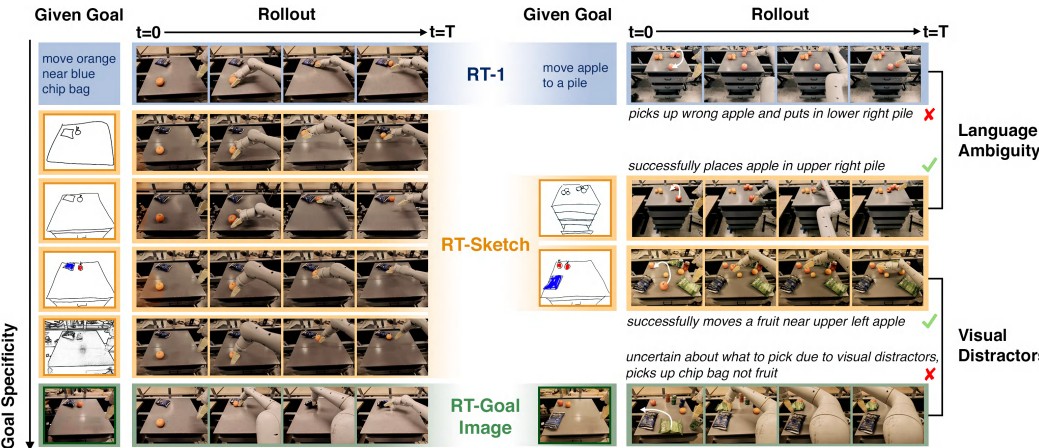

Figure 1: (Left) Qualitative rollouts comparing RT-Sketch, RT-1, and RT-Goal-Image, (right) highlighting RT-Sketch's robustness to (top) ambiguous language and (bottom) visual distractors.

~~Between natural language, which lacks granularity to unambiguously specify goals, and images, which overspecify goals in unnecessary detail, leading to the need for internet-scale data for generalization, we recognize that current frameworks lack a goal representation which adequately captures user intent in a convenient yet expressive manner.~~ While natural language is highly flexible, it can also be highly ambiguous or require lengthy descriptions. This quickly becomes difficult in long-horizon tasks or those requiring spatial awareness. Meanwhile, goal images over-specify goals in unnecessary detail, leading to the need for internet-scale data for generalization.

To ~~this end~~ address these challenges, we ~~study~~ *hand-drawn sketches* as a convenient yet expressive modality for goal specification in visual imitation learning. By virtue of being minimal, sketches are still easy for users to provide on the fly like language Yet unlike language, they (1) provide more, but allow for more spatial-aware task specification. Like goal images, sketches readily integrate with off-the-shelf policy architectures that take visual input, but provide an added level of goal abstraction that ignores unnecessary pixel-level details. Finally, the quality and selective inclusion/exclusion of details in a sketch can help a downstream policy distinguish task relevant from irrelevant details. ~~without needing to faithfully preserve pixel-level details as in an image, and (2) help a downstream policy disambiguate task-relevant from -irrelevant objects based on their selective inclusion, exclusion, or level of detail. Furthermore, sketches readily integrate with off-the-shelf policy architectures that take visual representations as input.~~

In this work, we present RT-Sketch, a goal-conditioned policy for manipulation that takes a user-provided hand-drawn sketch of the desired scene as input, and outputs actions. The novel architecture of RT-Sketch modifies the original RT-1 language-to-action Transformer architecture (Brohan et al., 2023b) to consume visual goals rather than language, allowing for flexible conditioning on sketches, images, or any other visually representable goals. To enable this, we concatenate a goal sketch and history of observations as input before tokenization, omitting language. We train RT-Sketch on a dataset of 80K trajectories paired with synthetically produced goal sketches, generated by an image-to-sketch stylization network trained from a few hundred image-sketch pairs.

We evaluate RT-Sketch across six manipulation skills on real robots involving tabletop object rearrangements on a countertop with drawers, subject to a wide range of scene variations. These skills include moving objects near to one another, knocking a can sideways, placing a can upright, closing a drawer, and opening a drawer. Experimentally, we find that RT-Sketch performs on a similar level to image or language-conditioned agents in straightforward settings. When language instructions are ambiguous, or in the presence of visual distractors, we find that RT-Sketch achieves $\sim 2X$ more spatial precision and alignment scores, as assessed by human labelers, over language or goal image-conditioned policies (see Fig. 1 (right)). Additionally, we show that RT-Sketch can handle different levels of input specificity, ranging from rough sketches to more scene-preserving, colored drawings (see Fig. 1 (left)).

## 2  RELATED WORK

In this section, we discuss prior methods for goal-conditioned imitation learning ~~which operate on traditional goal representations.~~ We also highlight ongoing efforts towards image-sketch conversion, which open new possibilities for goal-conditioning modalities which are underexplored in robotics.

**Goal-Conditioned Imitation Learning**   Despite the similarity in name, our learning of manipulation policies conditioned on hand-drawn sketches of the desired scene is different from the notion of policy sketches (Andreas et al., 2017), symbolic representations of task structure describing its subcomponents. Reinforcement learning (RL) is not easily applicable in our scenario, as it is nontrivial to define a reward objective which accurately quantifies alignment between a provided scene sketch and states visited by an agent during training. We instead focus on imitation learning (IL) techniques, particularly the goal-conditioned setting (Ding et al., 2019).

Goal-conditioned IL has proven useful in settings where a policy must be able to handle spatial or semantic variations for the same task (Argall et al., 2009). These settings include rearrangement of multiple objects (Brohan et al., 2023b;a; Lynch et al., 2023; Manuelli et al., 2019), kitting (Zakka et al., 2020), folding of deformable objects into different configurations (Ganapathi et al., 2021), and search for different target objects in clutter (Danielczuk et al., 2019). However, these approaches tend to either rely on language (Brohan et al., 2023b; Lynch & Sermanet, 2020; Lynch et al., 2023; Karamcheti et al., 2023; Shao et al., 2020), or goal images (Danielczuk et al., 2019) to specify variations. Follow-up works enable multimodal conditioning on either goal images and language (Jang et al., 2022), in-prompt images (Jiang et al., 2022), or image embeddings (Manuelli et al., 2019; Zakka et al., 2020; Ganapathi et al., 2021). However, all of these representations are ultimately derived from raw images or language in some way, which overlooks the potential for more abstract goal representations that are easy to specify but preserve spatial awareness, such as sketches.

In addition to their inflexibility in terms of goal representation, goal-conditioned IL tends to overfit to demonstration data and fails to handle even slight distribution shift in new scenarios (Ross et al., 2011). For language-conditioning, distribution shift can encompass semantic or spatial ambiguity, novel instructions or phrasing, or unseen objects (Jang et al., 2022; Brohan et al., 2023b). Goal-image conditioning is similarly susceptible to out-of-distribution visual shift, such as variations in lighting or object appearances, or unseen background textures (Burns et al., 2022; Belkhale et al., 2023). We instead opt for sketches which are minimal enough to combat visual distractors, yet expressive enough to provide unambiguous goals. Prior work, including (Barber et al., 2010) and (Porfirio et al., 2023), have shown the utility of sketches over pure language for navigation and limited manipulation settings. However, the sketches explored in these works are largely intended to guide low-level motion at the joint-level for manipulation, or provide explicit directional cues for navigation. Cui et al. (2022) considers sketches amongst other modalities as an input for goal-conditioned manipulation, but does not explicitly train a policy conditioned on sketches. They thus came to the conclusion that the scene image is better than the sketch image at goal specification. Our result is different and complementary, in that policies trained to take sketches as input outperform a scene image conditioned policy, by 1.63x and 1.5x in terms of Likert ratings for perceived spatial and semantic alignment, subject to visual distractors.

**Image-Sketch Conversion**   In recent years, sketches have gained increasing popularity within the computer vision community for applications such as object detection (Chowdhury et al., 2023a; Chowdhury et al., 2023; Chowdhury et al., 2022), visual question answering (Qiu et al., 2022; Qiu et al., 2023), and scene understanding (Chowdhury et al., 2023b), either in isolation or in addition to text and images. When considering how best to incorporate sketches in IL, an important design choice is whether to take sketches into account (1) at test time (i.e., converting a sketch to another goal modality compatible with a pre-trained policy), or (2) at training time (i.e., explicitly training an IL policy conditioned on sketches). For (1), one could first convert a given sketch to a goal image, and then roll out a vanilla goal-image conditioned policy. This could be based on existing frameworks for sketch-to-image conversion, such as ControlNet (Zhang & Agrawala, 2023), GAN-style approaches (Koley et al., 2023), or text-to-image synthesis, such as InstructPix2Pix (Brooks et al., 2023) or Stable Diffusion (Rombach et al., 2022). While these models produce photorealistic results under optimal conditions, they do not jointly handle image generation and style transfer, making it

unlikely for generated images to match the style of an agent observations. At the same time, these approaches are susceptible to producing hallucinated artifacts, introducing distribution shifts (Zhang & Agrawala, 2023).

Based on these challenges, we instead opt for (2), and consider image-to-sketch conversion techniques for hindsight relabeling of terminal images in pre-recorded demonstration trajectories. Recently, Vinker et al. (2022b;a) proposes networks for predicting Bezier curve-based sketches of input image objects or scenes. Sketch quality is supervised by a CLIP-based alignment metric. While these approaches generate sketches of high visual fidelity, test-time optimization takes on the order of minutes, which does not scale to the typical size of robot learning datasets (hundreds to thousands of demonstration trajectories). Meanwhile, conditional generative adversarial networks (cGANs) such as Pix2Pix (Isola et al., 2017) have proven useful for scalable image-to-image translation. Most related to our work is that of Li et al. (2019), which trains a Pix2Pix model to produce sketches from given images on a large crowd-sourced dataset of $5K$ paired images and line drawings. We build on this work to fine-tune an image-to-sketch model on robot trajectory data, and show its utility for enabling downstream manipulation from sketches.

## 3 SKETCH-CONDITIONED IMITATION LEARNING

In this section, we will first introduce our problem of learning a sketch-conditioned policy. We will then discuss our approach to train an end-to-end sketch-to-action IL agent. First, in Section 3.1, we discuss our instantiation of an auxiliary image-to-sketch translation network which automatically generates sketches from a reference image. In Section 3.2, we discuss how we use such a model to automatically hindsight relabel an existing dataset of demonstrations with synthetically generated goal sketches, and train a sketch-conditioned policy on this dataset.

**Problem Statement** Our goal is to learn a manipulation policy conditioned on a goal *sketch* of the desired scene state and a history of interactions. Formally, we denote such a policy by $\pi_{\text{sketch}}(a_t|g, \{o_j\}_{j=1}^t)$, where $a_t$ denotes an action at timestep $t$, $g \in \mathbb{R}^{W \times H \times 3}$ is a given goal sketch with width $W$ and height $H$, and $o_t \in \mathbb{R}^{W \times H \times 3}$ is an observation at time $t$. At inference time, the policy takes a given goal sketch along with a history of RGB image observations to infer an action to execute. In practice, we condition $\pi_{\text{sketch}}$ on a history of $D$ previous observations rather than all observations from the initial state at $t = 1$. To train such a policy, we assume access to a dataset $\mathcal{D}_{\text{sketch}} = \{g^n, \{(o_t^n, a_t^n)\}_{t=1}^{T^{(n)}}\}_{n=1}^N$ of $N$ successful demonstrations, where $T^{(n)}$ refers to the length of the $n^{th}$ trajectory in timesteps. Each episode of the dataset consists of a given goal sketch and a corresponding demonstration trajectory, with image observations recorded at each timestep. Our goal is to thus learn the sketch-conditioned imitation policy $\pi_{\text{sketch}}(a_t|g, \{o_j\}_{j=1}^t)$ trained on this dataset $\mathcal{D}_{\text{sketch}}$.

### 3.1 IMAGE-TO-SKETCH TRANSLATION

Training a sketch-conditioned policy requires a dataset of robot trajectories that are each paired with a sketch of the goal state achieved by the robot. Collecting such a dataset from scratch at scale, including the trajectories themselves and manually drawn sketches, can easily become impractical. Thus, we instead aim to learn an image-to-sketch translation network $\mathcal{T}(g|o)$ that takes an image observation $o$ and outputs the corresponding goal sketch $g$. This network can be used to post-process an existing dataset of demonstrations $\mathcal{D} = \{\{(o_t^n, a_t^n)\}_{t=1}^{T^{(n)}}\}_{n=1}^N$ with image observations by appending a synthetically generated goal sketch to each demonstration. This produces a dataset for sketch-based IL: $\mathcal{D}_{\text{sketch}} = \{g^n, \{(o_t^n, a_t^n)\}_{t=1}^{T^{(n)}}\}_{n=1}^N$.

**RT-1 Dataset** In this work, we rely on an existing dataset of visual demonstrations collected by prior work (Brohan et al., 2023b). RT-1 is a prior language-to-action imitation learning agent trained on a large-scale dataset ($80K$ trajectories) of VR-teleoperated demonstrations that include skills such as moving objects near one another, placing cans and bottles upright or sideways, opening and closing cabinets, and performing pick and place on countertops and drawers (Brohan et al., 2023b). Here, we repurpose the RT-1 dataset and further adapt the RT-1 policy architecture to accommodate sketches, detailed in Section 3.2.

**Assumptions on Sketches**   We acknowledge that there are innumerable ways for a human to provide a sketch corresponding to a given image of a scene. In this work, we make the following assumptions about input sketches for a controlled experimental validation procedure. In particular, we first assume that a given sketch respects the task-relevant contours of an associated image, such that tabletop edges, drawer handles, and task-relevant objects are included and discernible in the sketch. We do not assume contours in the sketch to be edge-aligned or pixel-aligned with those in an image. We do assume that the input sketch consists of black outlines at the very least, with shading in color being optional. We further assume that sketches do not contain information not present in the associated image, such as hallucinated objects, scribbles, or textual annotations, but may omit task-irrelevant details that appear in the original image.

**Sketch Dataset Generation**   To train an image-to-sketch translation network $\mathcal{T}$, we collect a new dataset $\mathcal{D}_{\mathcal{T}} = \{(o_i, g_i^1, \ldots, g_i^{L^{(i)}})\}_{i=1}^{M}$ consisting of $M$ image observations $o_i$ each paired with a set of goal sketches $g_i^1, \ldots, g_i^{L^{(i)}}$. Those represent $L^{(i)}$ *different* representations of the same image $o_i$, in order to account for the fact that there are multiple, valid ways of sketching the same scene. To collect $\mathcal{D}_{\mathcal{T}}$, we take 500 randomly sampled terminal images from demonstration trajectories in the RT-1 dataset, and manually draw sketches with black lines on a white background capturing the tabletop, drawers, and relevant objects visible on the manipulation surface. While we personally annotate each robot observation with a single sketch only, we add this data to an existing, much larger non-robotic dataset (Li et al., 2019). This dataset captures inter-sketch variation via multiple crowdsourced sketches per image. We do not include the robot arm in our manual sketches, as we find a minimal representation to be most natural. Empirically, we find that our policy can handle such sketches despite actual goal configurations likely having the arm in view. We collect these drawings using a custom digital stylus drawing interface in which a user draws an edge-aligned sketch over the original image (Appendix Fig. 15). The final recorded sketch includes the user's strokes in black on a white canvas with the original image dimensions.

**Image-to-Sketch Training**   We implement the image-to-sketch translation network $\mathcal{T}$ with the Pix2Pix conditional generative adversarial network (cGAN) architecture, which is composed of a generator $G_{\mathcal{T}}$ and a discriminator $D_{\mathcal{T}}$ (Isola et al., 2017). The generator $G_{\mathcal{T}}$ takes an input image $o$, a random noise vector $z$, and outputs a goal sketch $g$. The discriminator $D_{\mathcal{T}}$ is trained to discriminate amongst artificially generated sketches and ground truth goal sketches. We utilize the standard cGAN supervision loss to train both (Li et al., 2019; Isola et al., 2017):

$$\mathcal{L}_{\text{cGAN}} = \min_{G_{\mathcal{T}}} \max_{D_{\mathcal{T}}} \mathbb{E}_{o,g}[\log D_{\mathcal{T}}(o, g)] + \mathbb{E}_{o,g}[\log(1 - D_{\mathcal{T}}(o, G_{\mathcal{T}}(o, g)))] \tag{1}$$

We also add the $\mathcal{L}_1$ loss to encourage the produced sketches to align with the ground truth sketches as in (Li et al., 2019). To account for the fact that there may be multiple valid sketches for a given image, we only penalize the minimum $\mathcal{L}_1$ loss incurred across all $L^{(i)}$ sketches provided for a given image as in Li et al. (2019). This is to prevent wrongly penalizing $\mathcal{T}$ for producing a valid sketch that aligns well with one example but not another simply due to stylistic differences in the ground truth sketches. The final objective is then a $\lambda$-weighted combination of the average cGAN loss and the minimum alignment loss:

$$\mathcal{L}_{\mathcal{T}} = \frac{\lambda}{L^{(i)}} \sum_{k=1}^{L^{(i)}} \mathcal{L}_{\text{cGAN}}(o_i, g_i^{(k)}) + \min_{k \in \{1, \ldots, L^{(i)}\}} \mathcal{L}_1(o_i, g_i^{(k)}) \tag{2}$$

In practice, we supplement the 500 manually drawn sketches from $\mathcal{D}_{\mathcal{T}}$ by leveraging the existing larger-scale Contour Drawing Dataset (Li et al., 2019). We refer to this dataset as $\mathcal{D}_{\text{CD}}$, which contains 1000 examples of internet-scraped images containing objects, people, animals from Adobe Stock, paired with $L^{(i)} = 5$ crowd-sourced black and white outline drawings per image collected on Amazon Mechanical Turk. Visualizations of this dataset are provided in Appendix Fig. 4. We first take a pre-trained image-to-sketch translation network $\mathcal{T}_{\text{CD}}$ (Li et al., 2019) trained on $\mathcal{D}_{\text{CD}}$, with $L^{(i)} = 5$ sketches per image. Then, we fine-tune $\mathcal{T}_{\text{CD}}$ on $\mathcal{D}_{\mathcal{T}}$, with only $L^{(i)} = 1$ manually drawn sketch per robot observation, to obtain our final image-to-sketch network $\mathcal{T}$. Visualizations of the sketches generated by $\mathcal{T}$ for different robot observations are available in Fig. 5.

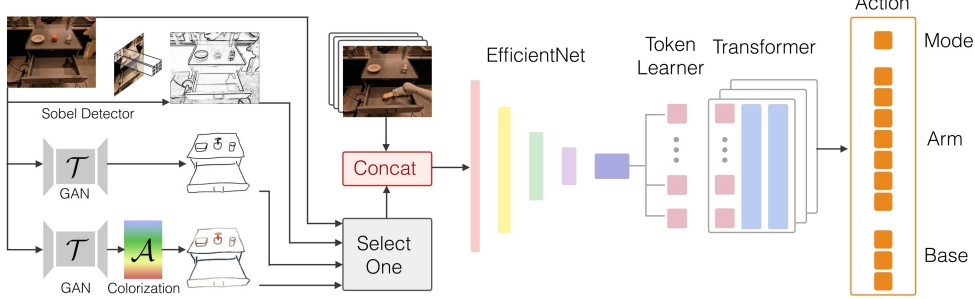

Figure 2: Architecture of RT-Sketch allowing different kinds of visual input. RT-Sketch adopts the Transformer (Vaswani et al., 2017) architecture with EfficientNet (Tan & Le, 2019) tokenization at the input, and outputs bucketized actions.

## 3.2 RT-SKETCH

With a means of translating image observations to black and white sketches via $\mathcal{T}$ (Section 3.1), we can automatically augment the existing RT-1 dataset with goal sketches. This results in a dataset, which we refer to as $\mathcal{D}_{\text{sketch}}$, which can be used for training our algorithm, RT-Sketch.

**RT-Sketch Dataset**   The original RT-1 dataset $\mathcal{D}_{\text{lang}} = \{i^n, \{(o_t^n, a_t^n)\}_{t=1}^{T^{(n)}}\}_{n=1}^N$ consists of $N$ episodes with a paired natural language instruction $i$ and demonstration trajectory $\{(o_t^n, a_t^n)\}_{t=1}^{T^n}$. We can automatically hindsight-relabel such a dataset with goal images instead of language goals (Andrychowicz et al., 2017). Let us denote the last step of a trajectory $n$ as $T^{(n)}$. Then the new dataset with image goals instead of language goals is $\mathcal{D}_{\text{img}} = \{o_{T^{(n)}}^n, \{(o_t^n, a_t^n)\}_{t=1}^{T^{(n)}}\}_{n=1}^N$, where we treat the last observation of the trajectory $o_{T^{(n)}}^n$ as the goal $g^n$. To produce a dataset for $\pi_{\text{sketch}}$, we can simply replace $o_{T^{(n)}}^n$ with $\hat{g}^n = \mathcal{T}(o_{T^{(n)}}^n)$ such that $\mathcal{D}_{\text{sketch}} = \{\hat{g}^n, \{(o_t^n, a_t^n)\}_{t=1}^{T^{(n)}}\}_{n=1}^N$.

To encourage the policy to afford different levels of input sketch specificity, we in practice produce goals by $\hat{g}^n = \mathcal{A}(o_{T^{(n)}}^n)$, where $\mathcal{A}$ is a randomized augmentation function. $\mathcal{A}$ chooses between simply applying $\mathcal{T}$, $\mathcal{T}$ with colorization during postprocessing (e.g., by superimposing a blurred version of the ground truth RGB image over the binary sketch), a classical Sobel operator (Sobel, 1968) for edge detection, or not applying any operators, which preserves the original ground truth goal image (Fig. 2). By co-training on all representations, we intend for RT-Sketch to handle a spectrum of specificity going from binary sketches; colorized sketches; edge detected images; and goal images (Appendix Fig. 5).

**RT-Sketch Model Architecture**   In our setting, we consider goals provided as sketches rather than language instructions as was done in RT-1. This change in the input representation necessitates a change in the model architecture. The original RT-1 policy relies on a Transformer architecture backbone (Vaswani et al., 2017). RT-1 first passes a history of $D = 6$ images through an EfficientNet-B3 model (Tan & Le, 2019) producing image embeddings, which are tokenized, and separately extracts textual embeddings and tokens via FiLM (Perez et al., 2018) and a Token Learner (Ryoo et al., 2021). The tokens are then fed into a Transformer which outputs bucketized actions. The output action dimensionality is 7 for the end-effector (x, y, z, roll, pitch, yaw, gripper width), 3 for the mobile base, (x, y, yaw), and 1 for a flag that can select amongst base movement, arm movement, and episode termination. To retrain the RT-1 architecture but accommodate the change in input representation, we omit the FiLM language tokenization altogether. Instead, we concatenate a given goal image or sketch with the history of images as input to EfficientNet, and extract tokens from its output, leaving the rest of the policy architecture unchanged. We visualize the RT-Sketch training inputs and policy architecture in Fig. 2. We refer to this architecture when trained only on images (i.e., an image goal-conditioned RT1 policy) as RT-Goal-Image and refer to it as RT-Sketch when it is trained on sketches as discussed in this section.

**Training RT-Sketch**   We can now train $\pi_{\text{sketch}}$ on $\mathcal{D}_{\pi_{\text{sketch}}}$ utilizing the same procedure as was used to train RT-1 (Brohan et al., 2023b), with the above architectural modifications. We fit $\pi_{\text{sketch}}$

using the behavioral cloning objective function. This aims to minimize the negative log-likelihood of an action provided the history of observations and a given sketch goal (Torabi et al., 2018):

$$J(\pi_{\text{sketch}}) = \sum_{n=1}^{N} \sum_{t=1}^{T^{(n)}} \log \pi_{\text{sketch}}(a_t^n | g^n, \{o_j\}_{j=1}^{t})$$

## 4 EXPERIMENTS

We seek to understand the ability of RT-Sketch to perform goal-conditioned manipulation as compared to policies that operate from higher-level goal abstractions like language, or more over-specified modalities, like goal images. To that end, we test the following four hypotheses:

**H1: RT-Sketch is successful at goal-conditioned IL.** While sketches are abstractions of real images, our hypothesis is that they are specific enough to provide manipulation goals to a policy. Therefore, we expect RT-Sketch to perform on a similar level to language goals (RT-1) or goal images (RT-Goal-Image) in straightforward manipulation settings.

**H2: RT-Sketch is able to handle varying levels of specificity.** There are as many ways to sketch a scene as there are people. Because we have trained RT-Sketch on sketches of varying levels of specificity, we expect it to be robust against variations of the input sketch for the same scene.

**H3: Sketches enable better robustness to distractors than goal images.** Sketches focus on task-relevant details of a scene. Therefore, we expect RT-Sketch to provide robustness against distractors in the environment that are not included in the sketch compared to RT-Goal-Image that operates on detailed image goals.

**H4: Sketches are favorable when language is ambiguous.** We expect RT-Sketch to provide a higher success rate compared to ambiguous language inputs when using RT-1.

### 4.1 EXPERIMENTAL SETUP

**Policies**    We compare RT-Sketch to the original language-conditioned agent RT-1 (Brohan et al., 2023b), and RT-Goal-Image, a policy identical in architecture to RT-Sketch, but taking a goal image as input rather than a sketch. All policies are trained on a multi-task dataset of $\sim 80$K real-world trajectories manually collected via VR teleoperation using the setup from Brohan et al. (2023b). These trajectories span a suite of common office and kitchen tasks such as picking and placing objects, reorienting cups and bottles upright or sideways, opening and closing drawers, and rearranging objects between drawers or a countertop.

**Evaluation protocol**    To ensure fair comparison, we control for the same initial and goal state of the environment across different policy rollouts via a catalog of well-defined evaluation scenarios that serve as references for human robot operators. For each scenario, we record an initial image (RGB observation) of the scene, the goal image (with objects manually rearranged as desired), a natural language task string describing the desired agent behavior to achieve the goal, and a set of hand-drawn sketches corresponding to the recorded goal image. At test time, a human operator retrieves a particular evaluation scenario from the catalog, aligns the physical robot and scene according to a reference image using a custom visualization utility, and places the relevant objects in their respective locations. Finally, the robot selects one of the goal representations (language, image, sketch, etc.) for the scenario as input to a policy. We record a video of the policy rollout for downstream evaluation (see Section 4.2). We perform all experiments using the Everyday Robot[2], which contains a mobile base, an overhead camera, and a 7-DoF manipulator arm with a parallel jaw gripper. All sketches for evaluation are collected with a custom manual drawing interface by a single human annotator on a tablet with a digital stylus.

**Performance Metrics**    Defining a standardized, automated evaluation protocol for goal alignment is non-trivial. Since binary task success is too coarse-grained and image-similarity metrics like frame-differencing or CLIP (Radford et al., 2021) tend to be brittle, we measure performance with

---

[2]everydayrobots.com

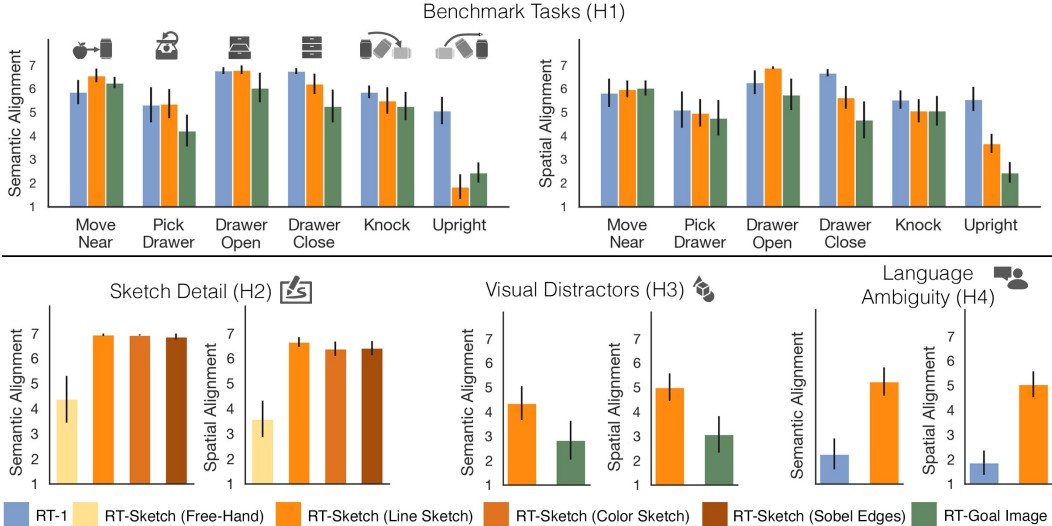

Figure 3: **Goal Alignment Results:** Average Likert scores for different policies rating perceived semantic alignment (**Q1**) and spatial alignment (**Q2**) to a provided goal. For straightforward benchmark manipulation tasks, RT-Sketch performs comparably and in some cases better than RT-1 and RT-Goal-Image in terms of both metrics, for 5 out of 6 skills (**H1**). RT-Sketch further exhibits the ability to handle sketches of different levels of detail (**H2**), while achieving better goal alignment than baselines when the visual scene is distracting (**H3**) or language would be ambiguous (**H4**). Error bars indicate standard error across labeler ratings.

two more targeted metrics. First, we quantify policy precision as the distance (in pixels) between object centroids in achieved and ground truth goal states, using manual keypoint annotations. Although leveraging out-of-the box object detectors to detect object centroids is a possibility, we want to avoid conflating errors in object detection (imprecise bounding box, wrong object, etc.) from manipulation error of the policy itself. Second, we gather human-provided assessments of perceived goal alignment, following the commonly-used Likert (Likert, 1932) rating scheme from 1 (Strongly Disagree) to 7 (Strongly Agree), for:

- (**Q1**) *The robot achieves **semantic alignment** with the given goal during the rollout.*
- (**Q2**) *The robot achieves **spatial alignment** with the given goal during the rollout.*

For **Q1**, we present labelers with the policy rollout video along with the given ground-truth language task description. We expect reasonably high ratings across all methods for straightforward manipulation scenarios (**H1**). Sketch-conditioned policies should yield higher scores than a language-conditioned policy when a task string is ambiguous (**H4**). **Q2** is instead geared at measuring to what degree a policy can spatially arrange objects as desired. For instance, a policy can achieve semantic alignment for the instruction *place can upright* as long as the can ends up in the right orientation. For **Q2**, we visualize a policy rollout side-by-side with a given visual goal (ground truth image, sketch, etc.) to assess perceived spatial alignment. We posit that all policies should receive high ratings for straightforward scenarios (**H1**), with a slight edge for visual-conditioned policies which implicitly have stronger spatial priors encoded in goals. We further expect that as the visual complexity of a scene increases, sketches may be able to better attend to pertinent aspects of a goal and achieve better spatial alignment than image-conditioned agents (**H3**), even for different levels of sketch specificity (**H4**). We provide a visualization of the assessment interface for **Q1** and **Q2** in Appendix Fig. 16. We note that we perform these human assessment surveys across 62 individuals (non-expert, unfamiliar with our system), where we assign between 8 and 12 people to evaluate each of the 6 different skills considered below.

## 4.2 EXPERIMENTAL RESULTS

In this section, we present our findings related to the hypotheses of Section 4. Tables 4.2 and 4.2 measure the spatial precision achieved by policies in terms of pixelwise distance, while Fig. 3 shows

the results of human-perceived semantic and spatial alignment, based on a 7-point Likert scale rating.

| Skill | Spatial Precision (RMSE in px.) | | | Failure Occurrence (Excessive Retrying) | | |
|---|---|---|---|---|---|---|
| | RT-1 | RT-Sketch | RT-Goal-Image | RT-1 | RT-Sketch | RT-Goal-Image |
| Move Near | $5.43 \pm 2.15$ | $\mathbf{3.49 \pm 1.38}$ | $3.89 \pm 1.16$ | $\mathbf{0.00}$ | 0.06 | 0.33 |
| Pick Drawer | $5.69 \pm 2.90$ | $4.77 \pm 2.78$ | $\mathbf{4.74 \pm 2.01}$ | $\mathbf{0.00}$ | 0.13 | 0.20 |
| Drawer Open | $4.51 \pm 1.55$ | $\mathbf{3.34 \pm 1.08}$ | $4.98 \pm 1.16$ | $\mathbf{0.00}$ | $\mathbf{0.00}$ | 0.07 |
| Drawer Close | $\mathbf{2.69 \pm 0.93}$ | $3.02 \pm 1.35$ | $3.71 \pm 1.67$ | $\mathbf{0.00}$ | $\mathbf{0.00}$ | 0.07 |
| Knock | $7.39 \pm 1.77$ | $\mathbf{5.36 \pm 2.74}$ | $5.63 \pm 2.60$ | $\mathbf{0.00}$ | 0.13 | 0.40 |
| Upright | $7.84 \pm 2.37$ | $5.08 \pm 2.08$ | $\mathbf{4.18 \pm 1.54}$ | 0.06 | $\mathbf{0.00}$ | 0.27 |
| Visual Distractors | - | $\mathbf{4.78 \pm 2.17}$ | $7.95 \pm 2.86$ | - | $\mathbf{0.13}$ | 0.67 |
| Language Ambiguity | $8.03 \pm 2.52$ | $\mathbf{4.45 \pm 1.54}$ | - | 0.40 | $\mathbf{0.13}$ | - |

Table 1: **Spatial Precision and Specific Failure Occurrence** : Left: We report the level of spatial precision achieved across policies, measured in terms of RMSE of the centroids of manipulated objects in achieved vs. given reference goal images. Darker shading indicates higher precision (lower centroid distance). Fig. 7 contains visualizations illustrating the degree of visual alignment that different RMSE values correspond to. Right: We report the proportion of rollouts in which different policies exhibit *excessive retrying* behavior. Bolded numbers indicate the most precise and least failure-prone policy for each skill.

**H1**: We evaluate 6 skills from the RT-1 benchmark (Brohan et al., 2023b): *move X near Y*, *place X upright*, *knock X over*, *open the X drawer*, *close the X drawer*, and *pick X from Y*. For each skill, we record 15 different catalog scenarios, varying both objects (16 unique in total) and their placements.

In general, we find that RT-Sketch performs on a comparable level to RT-1 and RT-Goal-Image for both semantic (**Q1**) and spatial alignment (**Q2**), achieving ratings in the 'Agree' to 'Strongly Agree' range on average for nearly all skills (Fig. 3 (top)). A notable exception is *upright*, where RT-Sketch essentially fails to accomplish the goal semantically (**Q1**), albeit with some degree of spatial alignment (**Q2**). Both RT-Sketch and RT-Goal-Image tend to position cans or bottles appropriately and then terminate, without realizing the need for reorientation (Appendix Fig. 8). This behavior results in low centroid-distance to the goal (darker gray in Table 4.2 (left)). RT-1, on the other hand, reorients cans and bottles successfully, but at the expense of higher error (Appendix Fig. 8, light color in Table 4.2 (left)). In our experiments, we also observe the occurrence of *excessive retrying behavior*, in which a policy attempts to align the current scene with a given goal with retrying actions such as grasping and placing. However, performing these low-level actions with a high degree of precision is challenging, and thus excessive retrying can actually disturb the scene leading to knocking objects off the table or undoing task progress. In Table 4.2, we report the proportion of rollouts in which we observe this behavior across all policies. We note that RT-Goal-Image is most susceptible to this failure mode, as a result of over-attending to pixel-level details and trying in excess to match a given goal exactly. Meanwhile, RT-Sketch and RT-1 are far less vulnerable, since both sketches and language provide a higher level of goal abstraction.

~~In this table, we further see that RT-Goal-Image has a tendency to over-attend to pixel-level details, which can result in excessive retrying behavior and failure to terminate when attempting to rearrange objects to exactly match a given goal image (darker gray in Section 4.2 (right), denoting more frequent failures).~~

**H2**: We next assess RT-Sketch's ability to handle input sketches of varied levels of detail (free-hand, edge-aligned line sketch, colorized line sketch, and a Sobel edge-detected image as an upper bound). Free-hand sketches are drawn with a reference image next to a blank canvas, while line sketches are drawn on a semi-transparent canvas overlaid on the image (see Appendix Fig. 15). We find such a UI to be convenient and practical, as an agent's current observations are typically available and provide helpful guides for sketching lines and edges. Across 5 trials each of the *move near* and *open drawer* skills, we see in Section 4.2 that all types of sketches produce reasonable levels of spatial precision. As expected, Sobel edges incur the least error, but even free-hand sketches, which do not necessarily preserve perspective projection, and line sketches, which are far sparser in detail, are not far behind. This is also reflected in the corresponding Likert ratings (Fig. 3 (left, bottom)). Free-hand sketches already garner moderate ratings (around 4) of perceived spatial and semantic alignment, but line sketches result in a marked performance improvement to nearly 7, on par with the upper bound of providing an edge-detected goal image. Adding color does not improve performance further, but leads to interesting qualitative differences in behavior (see Appendix Fig. 9).

| Skill | Free-Hand | Line Sketch | Color Sketch | Sobel Edges |
|---|---|---|---|---|
| Move Near | $7.21 \pm 2.76$ | $3.49 \pm 1.38$ | $3.45 \pm 1.03$ | $\mathbf{3.36 \pm 0.66}$ |
| Drawer Open | $3.75 \pm 1.63$ | $3.34 \pm 1.08$ | $2.48 \pm 0.50$ | $\mathbf{2.13 \pm 0.25}$ |

Table 2: **RT-Sketch Spatial Precision across Sketch Types (RMSE (centroid-distance) in px.** We report the spatial precision achieved by RT-Sketch subject to different input modalities. As expected, for less detailed and more rough sketches, RT-Sketch achieves lower precision (lighter shading), and for richer representations RT-Sketch is more precise (bolded, darker shading). Still, there is a relatively small difference in performance between line, color, and edge-detected representations, indicating RT-Sketch's ability to afford different levels of input specificity.)

**H3**: Next, we compare the robustness of RT-Sketch and RT-Goal-Image to the presence of visual distractors. We re-use 15 *move X near Y* trials from the catalog, but introducing $5 - 9$ distractor objects into the initial visual scene after alignment. This testing procedure is adapted from RT-1 generalization experiments referred to as *medium-high* difficulty (Brohan et al., 2023b). In Section 4.2 (left, bottom), we see that RT-Sketch exhibits far lower spatial errors on average, while producing higher semantic and spatial alignment scores over RT-Goal-Image( Fig. 3 (middle, bottom)). RT-Goal-Image is easily confused by the distribution shift introduced by distractor objects, and often cycles between picking up and putting down the wrong object. RT-Sketch, on the other hand, ignores task-irrelevant objects not captured in a sketch and completes the task in most cases (see Appendix Fig. 10).

**H4**: Finally, we evaluate whether sketches as a representation are favorable when language goals alone are ambiguous. We collect 15 scenarios encompassing 3 types of ambiguity in language instructions: instance ambiguity (**T1**) (e.g., *move apple near orange* when multiple orange instances are present), somewhat out-of-distribution (OOD) language (**T2**) (e.g., *move left apple near orange*), and highly OOD language (**T3**) (e.g., *complete the rainbow*) (see Appendix Fig. 11). While the latter two qualifications should intuitively help resolve ambiguities, they were not explicitly made part of the original RT-1 training (Brohan et al., 2023b), and hence only provide limited utility. In Section 4.2 (left, bottom), RT-Sketch achieves nearly half the error of RT-1, and a 2.39-fold and 2.79-fold score increase for semantic and spatial alignment, respectively (Fig. 3 (right, bottom)). For **T1** and **T2** scenarios, RT-1 often tries to pick up an instance of any object mentioned in the task string, but fails to make progress beyond that (Appendix Fig. 12). This further suggests the utility of sketches to express new, unseen goals with minimal overhead, when language could otherwise be opaque or difficult to express with only in-distribution vocabulary (Appendix Fig. 13).

**Limitations and Failure Modes** Firstly, the image-to-sketch generation network used in this work is fine-tuned on a dataset of sketches provided by a single human annotator, and we have yet to stress-test the generalization capabilities of RT-Sketch at scale with sketches produced by different people. Secondly, we note that RT-Sketch shows some inherent biases towards performing certain skills it was trained on, and occasionally performs the wrong skill. For a detailed breakdown of RT-Sketch's limitations and failure modes, please see Appendix C).

## 5 CONCLUSION

We propose RT-Sketch, a goal-conditioned policy for manipulation that takes a hand-drawn sketch of the desired scene as input, and outputs actions. To enable such a policy, we first develop a scalable way to generate paired sketch-trajectory training data via an image-to-sketch translation network, and modify the existing RT-1 architecture to take visual information as an input. Empirically, we show that RT-Sketch not only performs on a comparable level to existing language or goal-image conditioning policies for a number of manipulation skills, but is amenable to different degrees of sketch fidelity, and more robust to visual distractors or ambiguities. Future work will focus on extending hand-drawn sketches to more structured representations, like schematics or diagrams for assembly tasks. While powerful, sketches are not without their own limitations – namely ambiguity due to omittted details or poor quality sketches. In the future, we are excited by avenues for multimodal goal specification that can leverage the benefits of language, sketches, and other modalities to jointly resolve ambiguity from any single modality alone.

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

# RT-Sketch: Goal-Conditioned Imitation Learning from Hand-Drawn Sketches - Supplementary Material

In this section, we provide further details on the visual goal representations RT-Sketch sees at train and test time (Appendix A), qualitative visualizations of experimental rollouts (Appendix B), limitations (Appendix C) of RT-Sketch, as well as the interfaces used for data annotation, evaluation, and human assessment (Appendix D). To best understand RT-Sketch's performance and see its emergent capabilities, please refer to our website.[3]

## A  SKETCH GOAL REPRESENTATIONS

Since the main bottleneck to training a sketch-to-action policy like RT-Sketch is collecting a dataset of paired trajectories and goal sketches, we first train an image-to-sketch translation network $\mathcal{T}$ mapping image observations $o_i$ to sketch representations $g_i$, discussed in Section 3. To train $\mathcal{T}$, we first take a pre-trained network for sketch-to-image translation (Li et al., 2019) trained on the ContourDrawing dataset of paired images and edge-aligned sketches (Fig. 4). This dataset contains $L^{(i)} = 5$ crowdsourced sketches per image for 1000 images. By pre-training on this dataset, we hope to embed a strong prior in $\mathcal{T}$ and accelerate learning on our much smaller dataset. Next, we finetune $\mathcal{T}$ on a dataset of 500 manually drawn line sketches for RT-1 robot images. We visualize a few examples of our manually sketched goals in Fig. 5 under 'Line Drawings'.

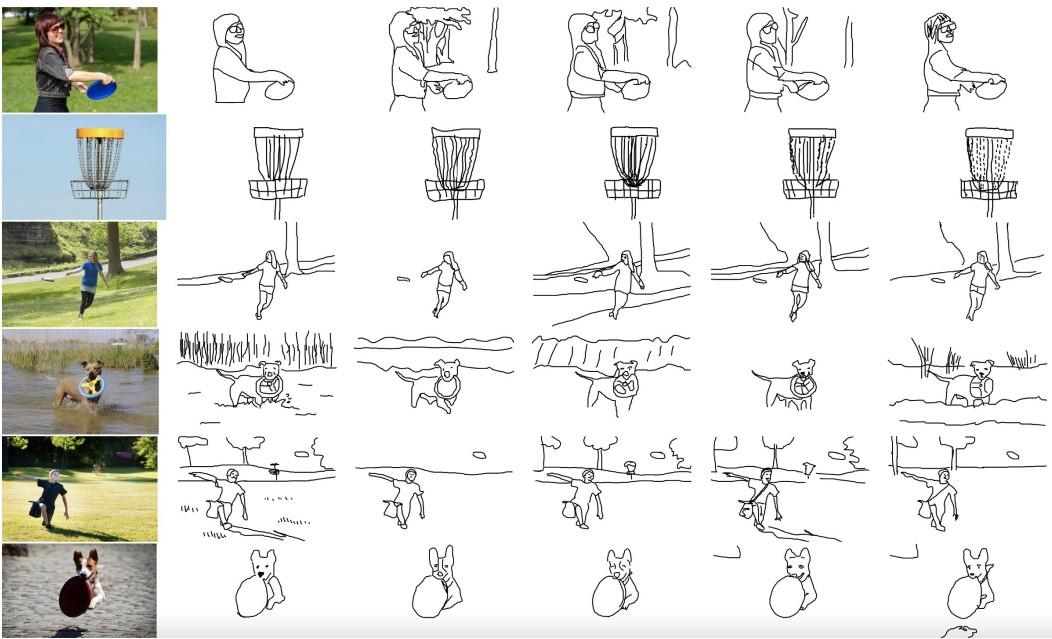

Figure 4: **ContourDrawing Dataset**: We visualize 6 samples from the ContourDrawing Dataset from (Li et al., 2019). For each image, 5 separate annotators provide an edge-aligned sketch of the scene by outlining on top of the original image. As depicted, annotators are encouraged to preserve main contours of the scene, but background details or fine-grained geometric details are often omitted. Li et al. (2019) then train an image-to-sketch translation network $\mathcal{T}$ with a loss that encourages aligning with at least one of the given reference sketches (Eq. (2)).

Notably, while we only train $\mathcal{T}$ to map an image to a black-and-white line sketch $\hat{g}_i$, we consider various augmentations $\mathcal{A}$ on top of generated goals to simulate sketches with varied colors, affine and perspective distortions, and levels of detail. Fig. 5 visualizes a few of these augmentations, such as automatically colorizing black-and-white sketches by superimposing a blurred version of the original RGB image, and treating an edge-detected version of the original image as a generated

---

[3]http://rt-sketch-anon.github.io

sketch to simulate sketches with a lot of details. We generate a dataset for training RT-Sketch by 'sketchifying' hind-sight relabeled goal images via $\mathcal{T}$ and $\mathcal{A}$.

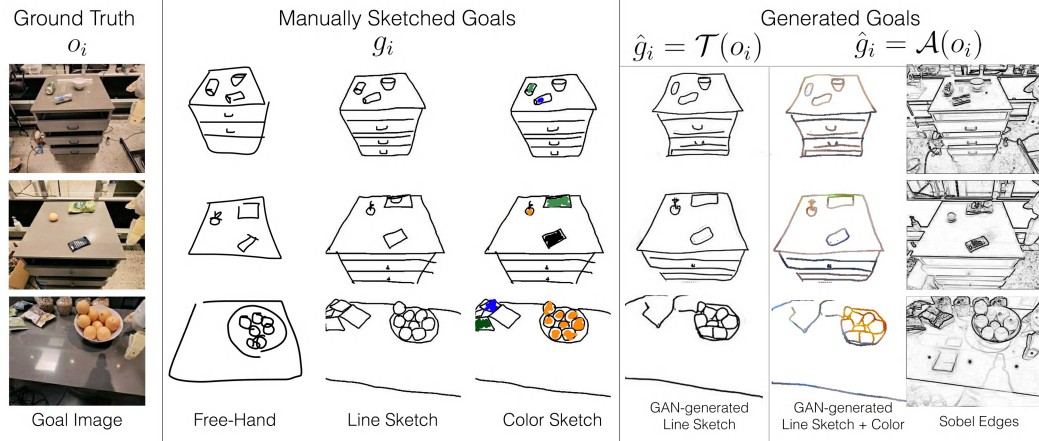

Figure 5: **Visual Goal Diversity**: RT-Sketch is capable of handling a variety of visual goals at both train and test time. RT-Sketch is trained on generated and augmented images like those shown on the right below 'Generated Goals'. But it can also interpret free-hand, line sketches, and colored sketches at test time such as those on the left below 'Manually Sketched Goals'.

Although RT-Sketch is only trained on generated line sketches, colorized line sketches, edge-detected images, and goal images, we find that it is able to handle sketches of even greater diversity. This includes non-edge aligned free-hand sketches and sketches with color infills, like those shown in Fig. 5.

## A.1 ALTERNATE IMAGE-TO-SKETCH TECHNIQUES

The choice of image-to-sketch technique we use is critical to the overall success of the RT-Sketch pipeline. We experiment with various other techniques before converging on the above approach. Recently, two recent works, CLIPasso (Vinker et al., 2022b) and CLIPAScene (Vinker et al., 2022a) explore methods for automatically generating a sketch from an image. These works pose sketch generation as inferring the parameters of Bezier curves representing "strokes" in order to produce a generated sketch with maximal CLIP-similarity to a given input image. These methods perform a per-image optimization to generate a plausible sketch, rather than a global batched operation across many images, limiting their scalability. Additionally, they are fundamentally more concerned with producing high-quality, aesthetically pleasing sketches which capture a lot of extraneous details.

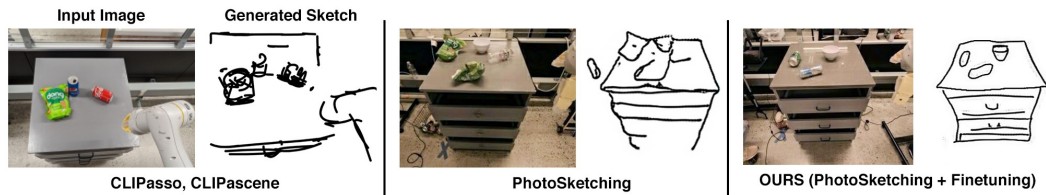

Figure 6: **Alternate Image-to-Sketch Techniques**

We, on the other hand, care about producing a minimal but reasonable-quality sketch. The second technique we explore is trying the pre-trained Photosketching GAN (Li et al., 2019) on internet data of paired images and sketches. However, this model output does not capture object details well, likely due to not having been trained on robot observations, and contains irrelevant sketch details. Finally, by finetuning this PhotoSketching GAN on our own data, the outputs are much closer to real, hand-drawn human sketches that capture salient object details as minimally as possible. We visualize these differences in Fig. 6.

# B ~~ROLLOUT~~ EVALUATION VISUALIZATIONS

To further interpret RT-Sketch's performance, we provide visualizations of the precision metrics and experimental rollouts. In Fig. 7, we visualize the degree of alignment RT-Sketch achieves, as quantified by the pixelwise distance of object centroids in achieved vs. given goal images. In Fig. 8, Fig. 9, Fig. 10, and Fig. 12, we visualize each policy's behavior for **H1, H2, H3** and **H4**, respectively. Fig. 11 visualizes the four tiers of difficulty in language ambiguity that we analyze for **H4**. To best understand RT-Sketch's performance, we kindly refer interested readers to our website containing a much more detailed overview and videos of all policies.

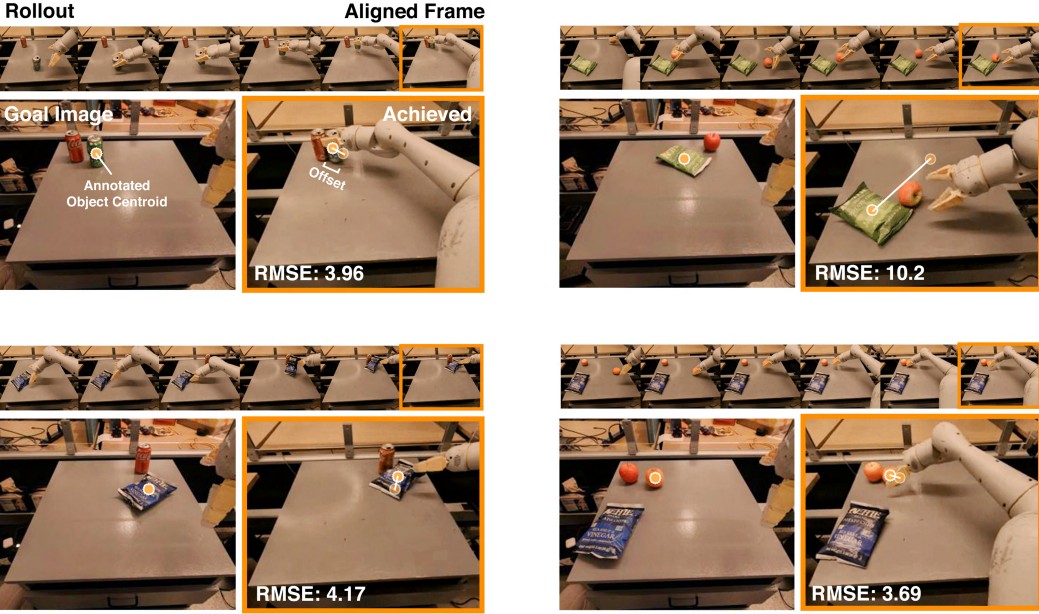

Figure 7: **Spatial Precision Visualization**: We visualize four trials of RT-Sketch on the Move Near skill, along with the measured spatial precision in terms of RMSE. To evaluate spatial precision, we have a human annotator annotate the frame that is visually most aligned, and then keypoints for the object that was moved in this frame and in the provided reference goal image. For each of the four trials, we visualize the rollout frames until alignment is achieved, along with the labeled object centroids and the offset in achieved vs. desired positions. The upper right example shows a failure of RT-Sketch in which the apple is moved instead of the chip bag, incurring a high RMSE. These visualizations are intended to better contextualize the numbers from Section 4.2.

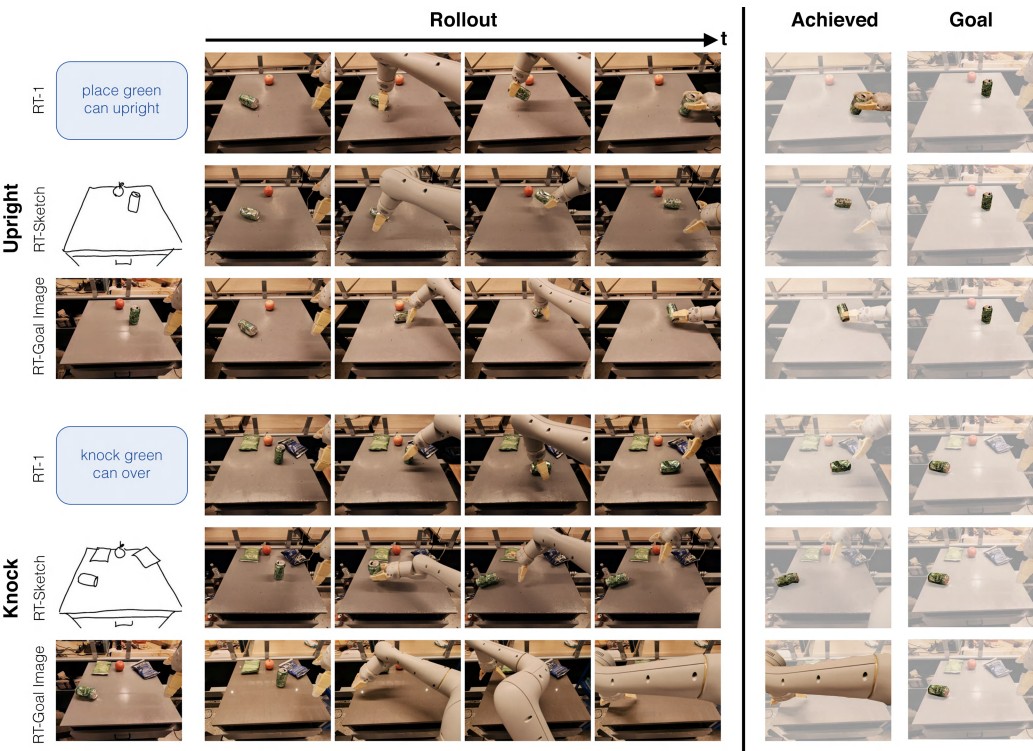

Figure 8: **H1 Rollout Visualization**: We visualize the performance of RT-1, RT-Sketch, and RT-Goal-Image on two skills from the RT-1 benchmark (*upright* and *knock*). For each skill, we visualize the goal provided as input to each policy, along with the policy rollout. We see that for both skills, RT-1 obeys the semantic task at hand by successfully placing the can upright or sideways, as intended. Meanwhile, RT-Sketch and RT-Goal-Image struggle with orienting the can upright, but successfuly knock it sideways. Interestingly, both RT-Sketch and RT-Goal-Image are able to place the can in the desired location (disregarding can orientation) whereas RT-1 does not pay attention to where in the scene the can should be placed. This is indicated by the discrepancy in position of the can in the achieved versus goal images on the right. This trend best explains the anomalous performance of RT-Sketch and RT-Goal-Image in perceived Likert ratings for the upright task (Fig. 3), but validates their comparably higher spatial precision compared to RT-1 across all benchmark skills (Section 4.2).

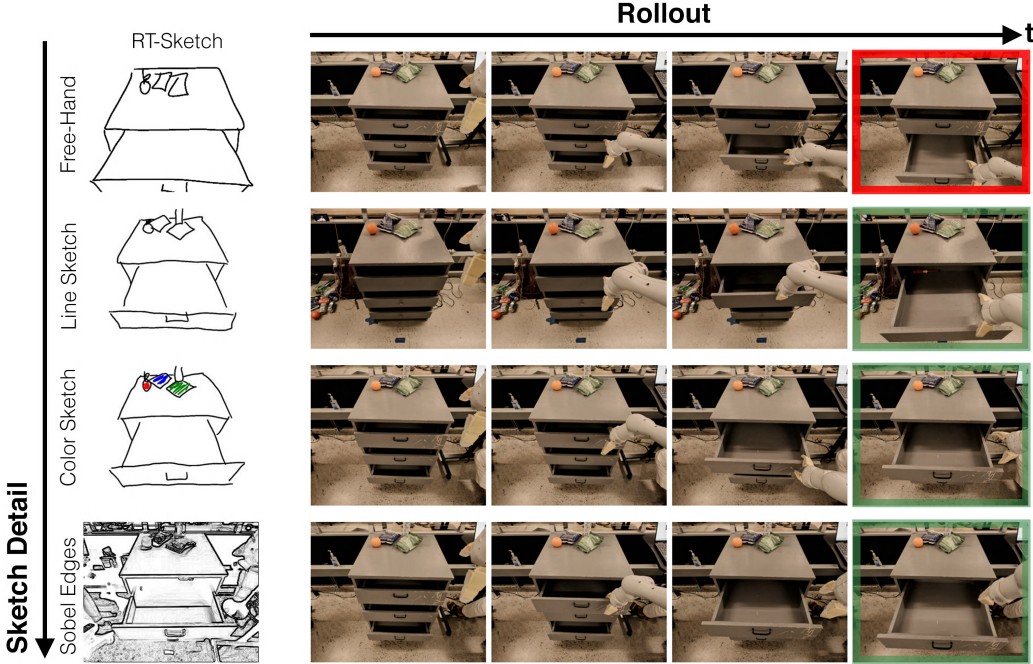

Figure 9: **H2 Rollout Visualization**: For the *open drawer* skill, we visualize four separate rollouts of RT-Sketch operating from different input types. Free-hand sketches are drawn without outlining over the original image, such that they can contain marked perspective differences, partially obscured objects (drawer handle), and roughly drawn object outlines. Line sketches are drawn on top of the original image using the sketching interface we present in Appendix Fig. 15. Color sketches merely add color infills to the previous modality, and Sobel Edges represent an upper bound in terms of unrealistic sketch detail. We see that RT-Sketch is able to successfully open the correct drawer for any sketch input except the free-hand sketch, without a noticeable performance gain or drop. For the free-hand sketch, RT-Sketch still recognizes the need for opening a drawer, but the differences in sketch perspective and scale can occasionally cause the policy to attend to the wrong drawer, as depicted.

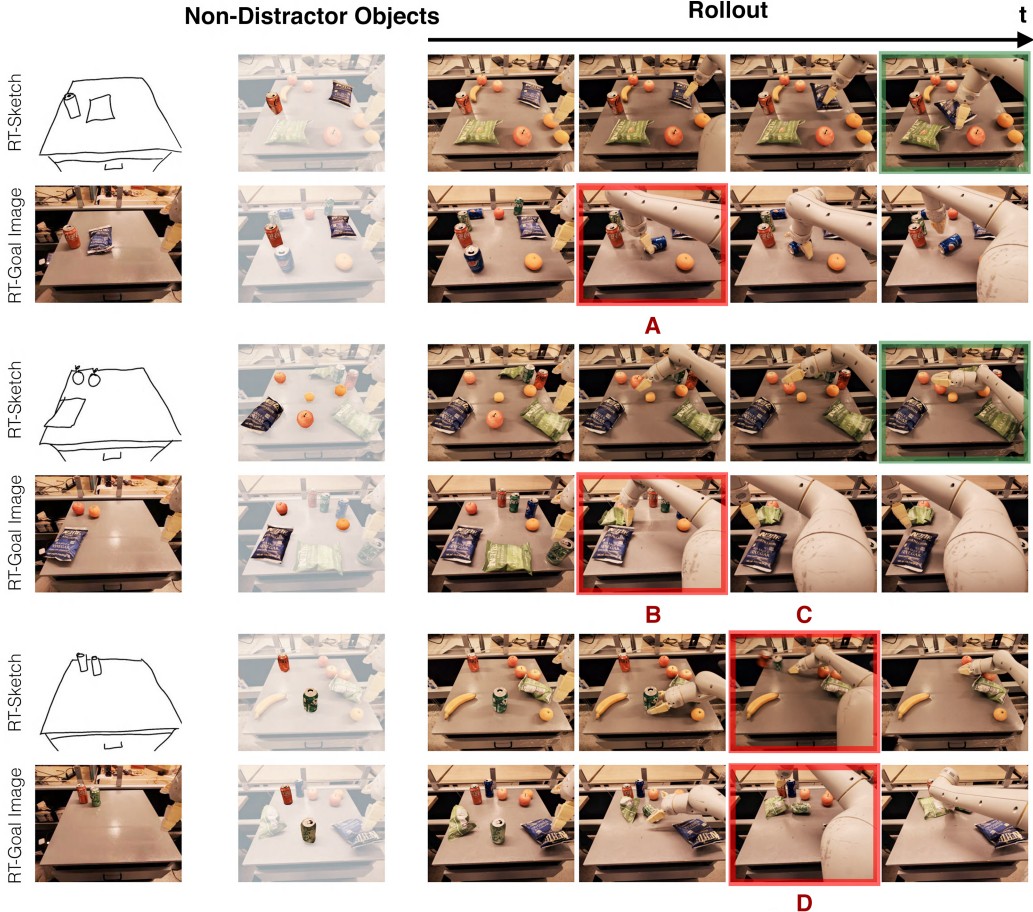

Figure 10: **H3 Rollout Visualization**: We visualize qualitative rollouts for RT-Sketch and RT-Goal-Image for 3 separate trials of the *move near* skill subject to distractor objects. In Column 2, we highlight the relevant non-distractor objects that the policy must manipulate in order to achieve the given goal. In Trial 1, we see that RT-Sketch successfuly attends to the relevant objects and moves the blue chip bag near the coke can. Meanwhile, RT-Goal-Image is confused about which blue object to manipulate, and picks up the blue pepsi can instead of the blue chip bag (A). In Trial 2, RT-Sketch successfully moves an apple near the fruit on the left. A benefit of sketches is their ability to capture instance multimodality, as any of the fruits highlighted in Column 2 are valid options to move, whereas this does not hold for an overspecified goal image. RT-Goal-Image erroneously picks up the green chip bag (B) instead of a fruit. Finally, Trial 3 shows a failure for both policies. While RT-Sketch successfully infers that the green can must be moved near the red one, it accidentally knocks over the red can (C) in the process. Meanwhile, RT-Goal-Image prematurely drops the green can and instead tries to pick the green chip bag (D).

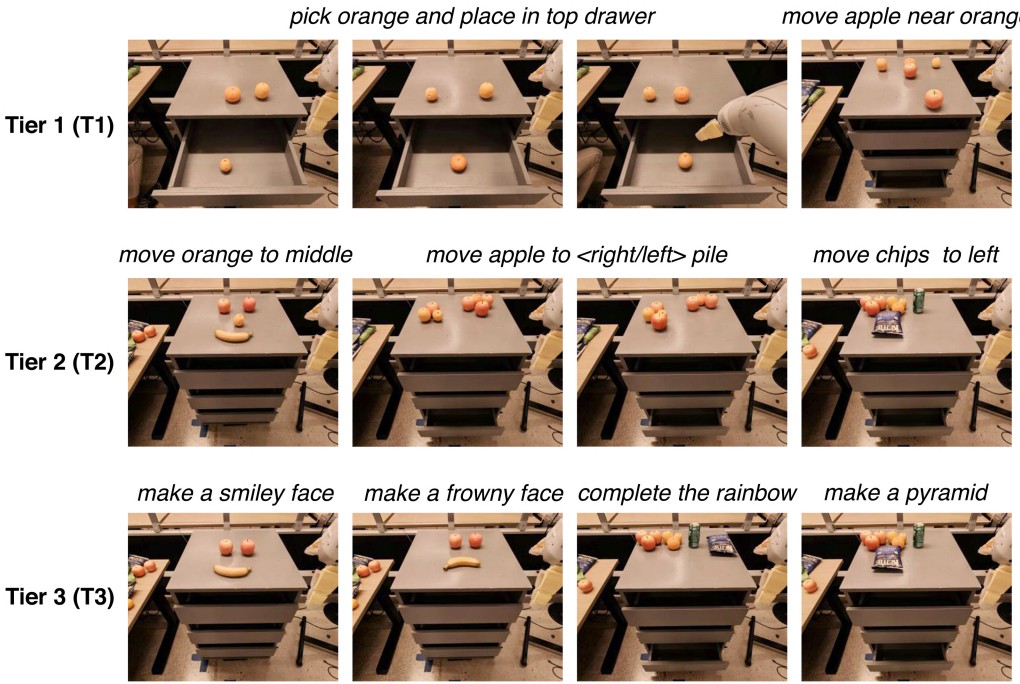

Figure 11: **H4 Tiers of Difficulty**: To test **H4**, we consider language instructions that are either ambiguous due the presence of multiple similar object instances (**T1**), are somewhat out-of-distribution for RT-1 (**T2**), or are far out-of-distribution and difficult to specify concretely without lengthier descriptions (**T3**). Each image represents the ground truth goal image paired with the task description.

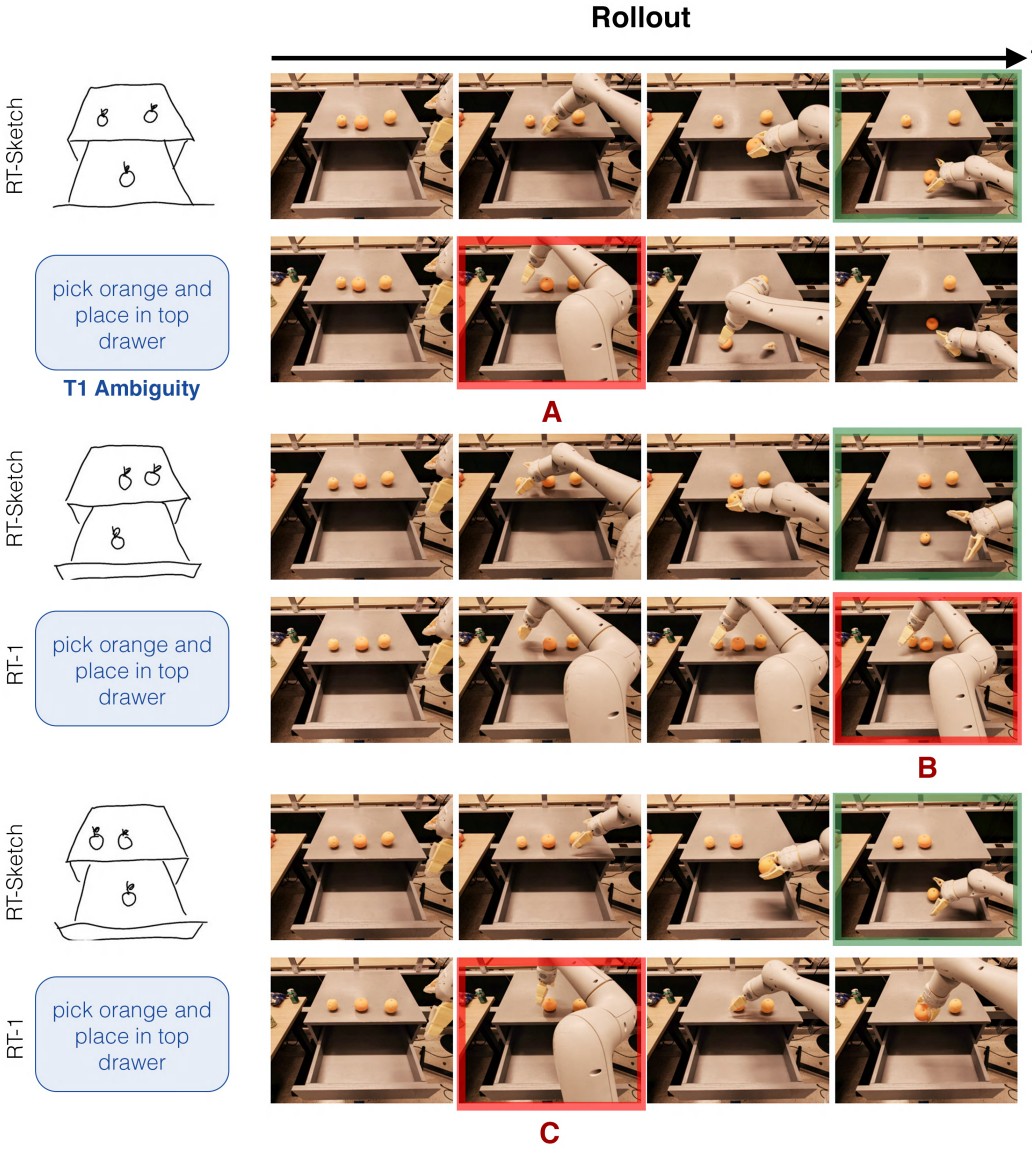

Figure 12: **H4 Rollout Visualization (T1 as visualized in Fig. 11)**: One source of ambiguity in language descriptions is mentioning an object for which there are multiple instances present. For example, we can easily illustrate three different desired placements of an orange in the drawer via a sketch, but an ambiguous instruction cannot easily specify which orange is relevant to pick and place. In all rollouts, RT-Sketch successfully places the correct orange in the drawer, while RT-1 either picks up the wrong object (A), fails to move to the place location (B), or knocks off one of the oranges (C). Although in this case, the correct orange to manipulate could easily be specified with a spatial relation like *pick up the ⟨ left/middle/right ⟩ orange*, we show below in Appendix Fig. 13 that this type of language is still out of the realm of RT-1's semantic familiarity.

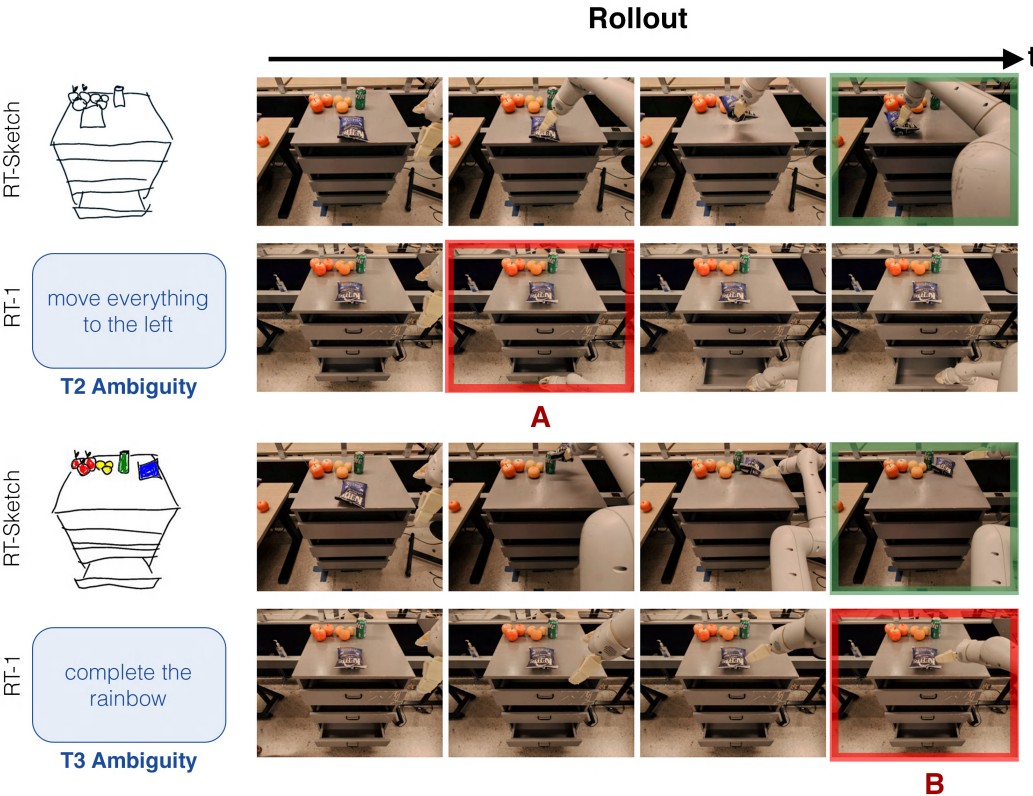

Figure 13: **H4 Rollout Visualization (T2-3 as visualized in Fig. 11)**: For **T2**, we consider language with spatial cues that intuitively should help the policy disambiguate in scenarios like the oranges in Fig. 12. However, we find that RT-1 is not trained to handle such spatial references, and this kind of language causes a large distribution shift leading to unwanted behavior. Thus, for the top rollout of trying to move the chip bag to the left where there is an existing pile, RT-Sketch completes the skill without issues, but RT-1 attempts to open the drawer instead of even attempting to rearrange anything on the countertop (A). For **T3**, we consider language goals that are even more abstract in interpretation, without explicit objects mentioned or spatial cues. Here, sketches are advantageous in their ability to succinctly communicate goals (i.e. visual representation of a rainbow), whereas the corresponding language task string is far too underspecified and OOD for the policy to handle (B).

## C    RT-SKETCH FAILURE MODES AND LIMITATIONS

While RT-Sketch  is performant at several manipulation benchmark skills, capable of handling different levels of sketch detail, robust to visual distractors, and unaffected by ambiguous language, it is not without failures and limitations.

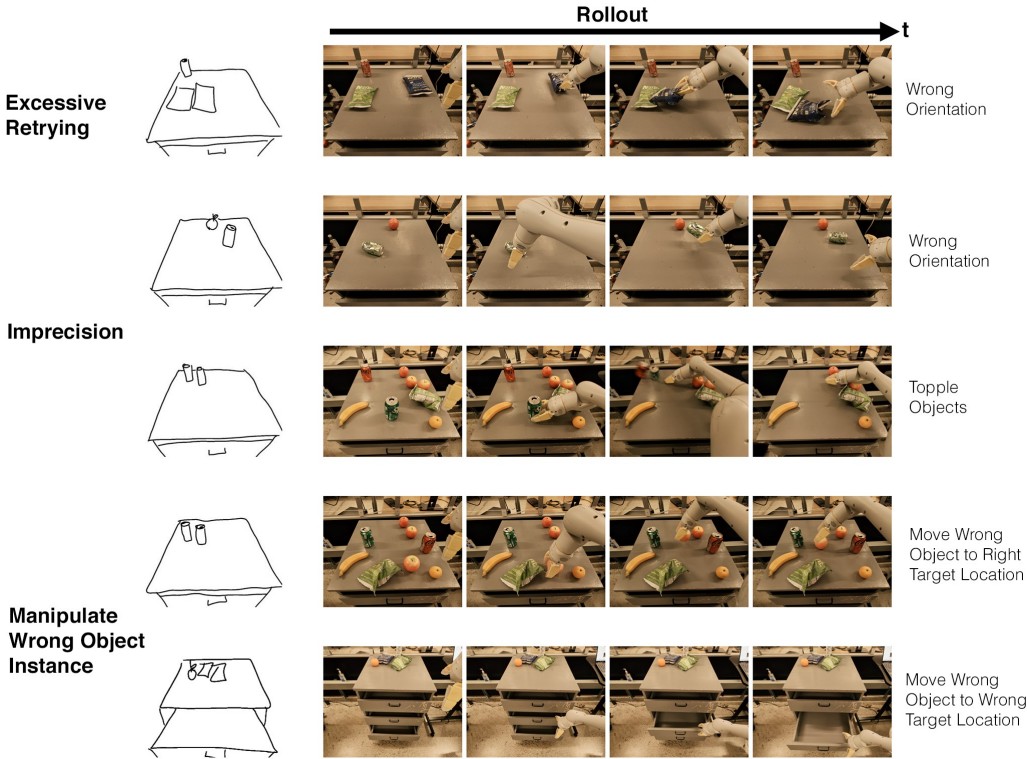

Figure 14: **RT-Sketch  Failure Modes**

In Fig. 14, we visualize the failure modes of RT-Sketch. One failure mode we see with RT-Sketch  is occasionally re-trying excessively, as a result of trying to align the scene as closely as possible. For instance, in the top row, Rollout Image 3, the scene is already well-aligned, but RT-Sketch keeps shifting the chip bag which causes some misalignment in terms of the chip bag orientation. Still, this kind of failure is most common with RT-Goal-Image (Section 4.2), and is not nearly as frequent for RT-Sketch. We posit that this could be due to the fact that sketches enable high-level spatial reasoning without over-attending to pixel-level details.

One consequence of spatial reasoning at such a high level, though, is an occasional lack of precision. This is noticeable when RT-Sketch orients items incorrectly (second row) or positions them slightly off, possibly disturbing other items in the scene (third row). This may be due to the fact that sketches are inherently imperfect, which makes it difficult to reason with such high precision.

Finally, we see that RT-Sketch occasionally manipulates the wrong object (rows 4 and 5). Interestingly, we see that a fairly frequent pattern of behavior is to manipulate the wrong object (orange in row 4) to the right target location (near green can in row 4). This may be due to the fact that the sketch-generating GAN has occasionally hallucinated artifacts or geometric details missing from the actual objects. Having been trained on some examples like these, RT-Sketch can mistakenly perceive the wrong object to be aligned with an object drawn in the sketch. However, the sketch still indicates the relative desired spatial positioning of objects in the scene, so in this case RT-Sketch still attempts to align the incorrect object with the proper place.

Finally, the least frequent failure mode is manipulating the wrong object to the wrong target location (i.e. opening the wrong drawer handle). This is most frequent when the input is a free-hand sketch, and could be mmitigated by increasing sketch detail (Section 4.2).

# D    EVALUATION AND ASSESSMENT INTERFACES

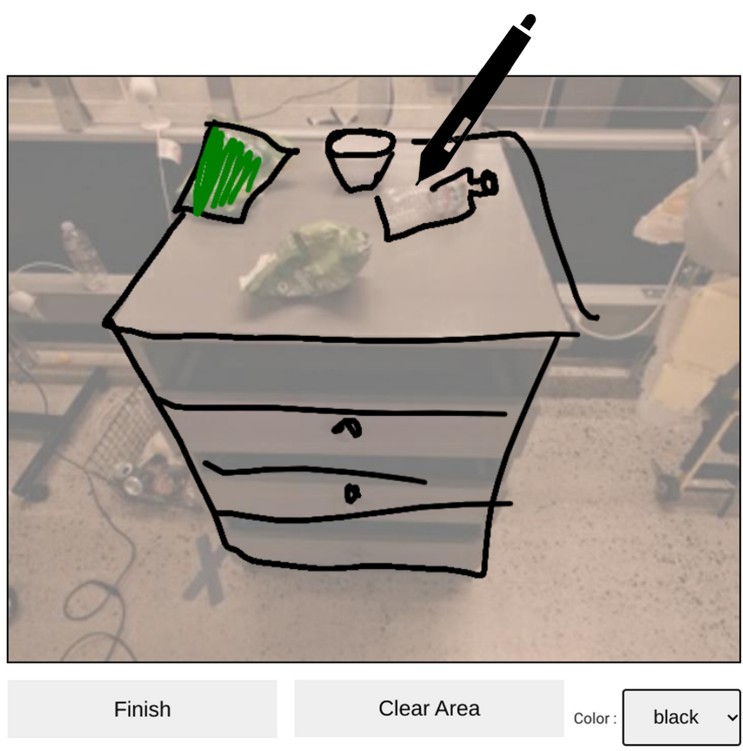

Figure 15: **Sketching UI**: We design a custom sketching interface for manually collecting paired robot images and sketches with which to train $\mathcal{T}$, and for sketching goals for evaluation. The interface visualizes the current robot observation, and provides the ability to draw on a digital screen with a stylus. The interface supports different colors and erasure. We note that intuitively, drawing on top of the image is not an unreasonable assumption to make, since current agent observations are far more readily available than a goal image, for instance. Additionally, the overlay is intended to make the sketching interface easy for the user to provide, without having to eyeball edges for the drawers or handles blindly.

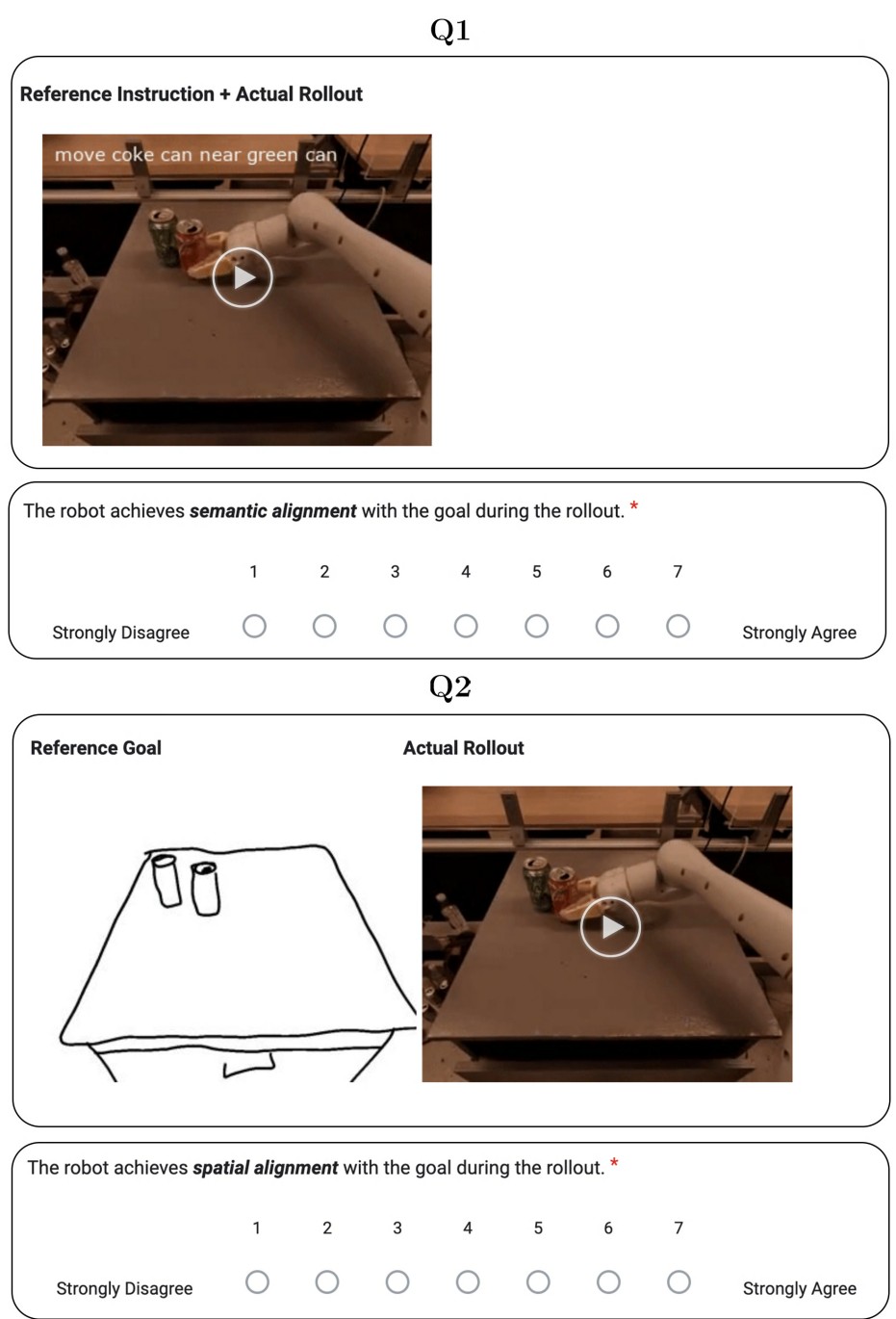

Figure 16: **Assessment UI**: For all skills and methods, we ask labelers to assess semantic and spatial alignment of the recorded rollout relative to the ground truth semantic instruction and visual goal. We show the interface above, where labelers are randomly assigned to skills and methods (anonymized). The results of these surveys are reported in Fig. 3.

