# OpenReview forum: "RT-Sketch: Goal-Conditioned Imitation Learning from Hand-Drawn Sketches"
_ICLR.cc/2024/Conference — Submitted to ICLR 2024_

### Official Review · Reviewer_rbwp · 2023-10-26

**Soundness:** 3 good
**Presentation:** 3 good
**Contribution:** 1 poor
**Rating:** 5
**Confidence:** 4

**Summary:**

This paper introduces an innovative approach to directing end-to-end robot manipulation tasks using sketches. The proposed model, referred to as RT-Sketch, interprets sketches of varying specificity, processes current and previous visual states, and predicts the corresponding robot actions. To obtain training data, a unique image-to-sketch model is utilized to convert terminal images from the Robot Transformer-1 (RT-1) into sketches, leading to the creation of the RT-Sketch dataset.

**Strengths:**

The proposed task is novel and interesting, potentially enhancing the efficiency of human-robot interactions through sketches. RT-Sketch can interpret and act upon sketches with varied levels of specificity, which suggests a degree of flexibility and adaptability in the model. The thorough experimental work effectively demonstrates the system's proficiency in executing the tasks assigned and its performance with the specific robot used.

**Weaknesses:**

##

1. The model, like its predecessor, the Robot Transformer (RT), remains an end-to-end agent. Although it showcases impressive performance, task comprehension is tied to a specific robot, hindering the system's ability to undertake out-of-domain tasks or generalize across various robots without necessitating retraining.
2. In terms of communication, a sketch represents more than just an enhanced image; it abstracts visual information and fosters the emergence of graphical conventions to boost efficiency (Qiu et al., 2022; Chen et al., 2023). However, the sketches in this work, even at the lowest specificity, seem to be merely processed images rather than abstractions, making this work more like an augmented version of RT-1 to sketch images.

The above limitations, in my opinion, restrict the work's contribution to addressing fundamental issues in human-robot collaboration and robot manipulation.

References:

- Qiu, S., Xie, S., Fan, L., Gao, T., Joo, J., Zhu, S. C., & Zhu, Y. (2022). Emergent graphical conventions in a visual communication game. *Advances in Neural Information Processing Systems*, *35*, 13119-13131.
- Lei, Z., Zhang, Y., Xiong, Y., & Chen, S. (2023). Emergent Communication in Interactive Sketch Question Answering. *arXiv [Cs.AI]*. Retrieved from http://arxiv.org/abs/2310.15597

**Questions:**

1. How does the model handle variations in sketch quality or style? Are there specific requirements for the sketches used to instruct the model?
2. I wonder if you can consider potential benefits of combining sketch and text descriptions to enhance task specification and promote more effective collaboration?

---

> ### Author Response · Authors · 2023-11-18
> **Response to Reviewer rbwp**
>
> Thank you for recognizing the novelty of sketches, the degree of flexibility they provide, and our policy’s performance! We appreciate your suggestions for improvement, addressed below.
>
> ***
> We have added your suggested references to the updated draft in related work, thanks a lot!
> ***
>
> > The model, like its predecessor, the Robot Transformer (RT), remains an end-to-end agent. Although it showcases impressive performance, task comprehension is tied to a specific robot, hindering the system's ability to undertake out-of-domain tasks or generalize across various robots without necessitating retraining.
>
> While we implement and evaluate our approach on a particular robot platform, we would like to emphasize that there is nothing in principle preventing our approach from applying to other robot embodiments and manipulation tasks more broadly. The primary contribution of this work is a framework for training a sketch-conditioned IL agent, without specific assumptions on the robot platform or even IL algorithm used. We are currently working on extending this platform to a Franka Panda robot by collecting demonstrations for a rearrangement task, and training the image-to-sketch GAN on a few paired image-sketch examples.
>
> Given recent efforts around crowdsourcing robot datasets [1], our image-to-sketch network could also certainly be trained across different robot embodiment images and paired sketches. This general-purpose image-to-sketch translation network could then be used to post-process many manipulation datasets in a hindsight fashion, enabling sketch-IL for free on different robots without any additional demo collection.
>
> ***
> > In terms of communication, a sketch represents more than just an enhanced image; it abstracts visual information and fosters the emergence of graphical conventions to boost efficiency (Qiu et al., 2022; Chen et al., 2023). However, the sketches in this work, even at the lowest specificity, seem to be merely processed images rather than abstractions, making this work more like an augmented version of RT-1 to sketch images.
>
> We would like to clarify that although we train on GAN-generated sketches (amidst other augmentations), **we evaluate RT-Sketch on real, hand-drawn sketches at test-time across all experiments.** We also come up with a principled way to evaluate whether RT-Sketch can handle sketches with varied levels of input specificity, ranging from free-hand drawings to line sketches to sketches with shading, all of which are drawn by hand at test-time rather than GAN-generated.
>
> ***
> > How does the model handle variations in sketch quality or style? Are there specific requirements for the sketches used to instruct the model?
>
> While the experiments in this work are based on sketches drawn by one human annotator, you bring up an extremely interesting point regarding how well RT-Sketch generalizes to sketches drawn by different individuals, with very different styles and even poor quality sketches. **We have since added [new experiments](https://rt-sketch-anon.github.io/additional.html#generalization) where we crowdsource 30 sketches across 6 human annotators, and evaluate perceived RT-Sketch’s spatial alignment via Likert ratings from 22 individuals.** RT-Sketch achieves high alignment scores on average with little drop-off between sketches drawn by different users, indicating its ability to interpret different human drawings and validating that our model is not overfit to a specific kind of sketch.
>
> ***
>
> > I wonder if you can consider potential benefits of combining sketch and text descriptions to enhance task specification and promote more effective collaboration?
>
> This is a great suggestion, and is an avenue of future work we are very excited about. As an initial exploration, **we have since trained a sketch-and-language-conditioned model with results [available on the website](https://rt-sketch-anon.github.io/additional.html#multimodal).** We find that conditioning on both performs on par with conditioning with either modality alone, and in some cases can address the ambiguity caused by one modality alone (i.e. sketches provide more spatial awareness than language, but language can often help the policy perform reorientation tasks which were previously difficult to express with a sketch alone). It would be very interesting to see if we could combine language more directly with a sketch, such as through scribbles or labels, or perform interactive reasoning where we elicit user feedback in the form of sketching or language to disambiguate goals.
>
> ***
> [1] Padalkar, Abhishek, et al. "Open x-embodiment: Robotic learning datasets and rt-x models." arXiv preprint arXiv:2310.08864 (2023).

---

> > ### Author Response · Authors · 2023-11-21
> > **Follow-Up to Reviewer rbwp**
> >
> > With the rebuttal period coming to a close, we hope we've addressed your concerns. We kindly ask that you consider raising your rating based on the above response. Feel free to let us know if you'd like us to address anything further. Thanks again for your thoughtful feedback!

---

### Official Review · Reviewer_GDgQ · 2023-10-27

**Soundness:** 2 fair
**Presentation:** 2 fair
**Contribution:** 2 fair
**Rating:** 3
**Confidence:** 3

**Summary:**

This work addresses a challenge in how humans specify the goal for robotics tasks. In related works, typically a goal-image or a natural language goal is given. The authors highlight the disadvantages that using these goals can cause. The authors propose another route, using hand-drawn sketches as a goal for the robotics task. RT-Sketch is proposed as an extension to RT1 as a goal-conditioned model to solve manipulation tasks when given a sketch of the final goal state. Experiments are first performed via a survey by using the Likert scores for perceived semantic and spatial alignment. After this table-top manipulation experiments are performed across 6 skills and ablations on visual distractors and language ambiguity.

**Strengths:**

- The direction the authors are going with this work, thinking about different ways of specifying a goal, will be useful for improving accessibility and has intellectual merits.
- The authors take steps to perform surveys to study human preference in specifying goals rather than just quantitative analysis through robot performance.
- Creation of the RT-Sketch dataset and the Image-To-Sketch model have technical merits.
- The limitations and failure modes are honest from a methodology perspective and are appreciated.
- Overall this paper is well written, Figure 2 does a good job of demonstrating the architecture of RT-Sketch and Figure 3 does a great job of conveying the survey results.

**Weaknesses:**

The main concerns with this manuscript come from two sources. The first is with respect to the motivation of using sketches to specify a goal and the second is concerning the quantitative robotics results.
## Motivation
Overall the reviewer is not convinced that the examples and arguments given motivate the superiority of using a sketch as a goal over using natural language.
- Regarding the granularity argument that the authors use in the introduction including the examples of "put utensils, ..., on the table" and "put the fork 2cm to the right...". Doesn't this demonstrate the flexibility that language has as a goal? Even if a human had to communicate the placement of utensils on a table, this would still seem easier than drawing a corresponding representation for an entire table.
- While language can be ambiguous, so can sketches. As an example, if a sketch was given with an empty table, am I telling the agent that I want it to throw out the garbage on the table or am I telling it to ignore the garbage? It would seem like the desirable solution would be to create more intelligent agents to create reasonable solutions or ask for clarifications when given an ambiguous problem.

## Robotics Results
- The metrics, "Spatial Precision" and "Failure Occurrence" are not carefully defined or motivated. Failure occurrence is not defined at all. The spatial precision metric is at best defined as "the distance (in pixels) between object centroids in achieved and ground truth goal states, using manual keypoint annotation". However, how these centroids are obtained and why manual keypoint annotation is necessary over using an off-the-shelf image classifier should have been mentioned.
- It is not obvious how big of a difference the errors are when looking at the RMSE in pixels. Could the errors when finding the centroid and the manual keypoint annotations be an issue in measuring this? Can a visualization be created to show this?
- Typically bolded numbers in a column/row represent the method with best performance. However, in the column for failure occurrences, RT-Goal-Image is bolded despite having the highest failure occurrence.
- The meaning of the shading of the cells of Table 1 adds a lot of confusion and should have been defined in the caption to improve readability. This confusion comes because, in one portion of the table, darker gray colors represent lower centroid distance and in another portion it represents the frequency of failures. This frequency of failure metric is also not well defined, how is this different than failure occurrence?

**Questions:**

Beyond the concerns and questions given in the weakness section.

- How many people took the survey? How was this survey conducted?

---

> ### Author Response · Authors · 2023-11-18
> **Response to Reviewer GDgQ**
>
> Thank you so much for your detailed review and suggestions for improvement! We hope the responses below, along with our updated writing and experiments, address your concerns.
> ***
> > Not convinced that the examples and arguments given motivate the superiority of using a sketch as a goal over using natural language.
>
> We apologize if the current framing makes it seem as though we view sketches as being strictly favorable to language in all cases. While we point out situations in which language alone might be ambiguous and sketches have a potential for allowing more spatial-aware task specification, this is not to say that language should be replaced by sketches altogether. We have made changes to the introduction and conclusion to better contextualize this work with this in mind.
>
> Regarding what scenarios language alone may be inconvenient, and where a sketch can serve as a meaningful alternative or complementary modality, we have since added [a section on our website](https://rt-sketch-anon.github.io/additional.html#why_sketches) with motivating examples.
>
> Our focus on sketches is largely an exploration around pushing the boundaries of goal specification more broadly beyond traditional modalities like language, and the design choices we can make to enable this at all. In the future, we are certainly excited about multimodal goal specification, and as a step towards this, we have since implemented a sketch-and-language-conditioned model which performs on par with, and better in some cases, than either modality alone. Specifically, in the reorientation task which our sketch-alone policy initially struggled with, we observe that sketches alleviate spatial ambiguity introduced by language, while language can help specify that reorientation (over just repositioning) is needed. Please refer to the [Additional Experiments section](https://rt-sketch-anon.github.io/additional.html#multimodal) on the website for more details. This result further suggests that sketches can powerfully complement other modalities.
> ***
> > Regarding the granularity argument that the authors use in the introduction including the examples of "put utensils, ..., on the table" and "put the fork 2cm to the right...". Doesn't this demonstrate the flexibility that language has as a goal? Even if a human had to communicate the placement of utensils on a table, this would still seem easier than drawing a corresponding representation for an entire table.
>
> We agree that these examples do illustrate the flexibility language offers, but we retain that having so much flexibility can also become highly inconvenient as the complexity of a task scales or the degree of spatial precision matters. For instance, in the given example of setting a table with multiple plates, utensils, napkins, decorations, etc., it would seem very inconvenient for a user to have to provide an instruction for the placement of every single object, along with such fine-grained corrections. On the other hand, a very long-winded description that captures all the placements of the objects in one shot would be difficult for any language-conditioned agent to interpret, let alone for someone to describe. Even an imperfect sketch of the objects and their relative placements could be quick to provide, and provide much more spatial context than a sentence for the policy to be able to roughly place objects.
>
> We have also shown in our experiments with visual distractors that sketches are capable of attending only to the objects we care about manipulating. Here, a human need not draw the entire table in one shot, but can use partial sketches to indicate subgoals in a long-horizon sequence. We fully agree that viewing these modalities in isolation is not necessarily the answer; a combination of a rough sketch, and verbal corrections, for instance, can go a long way in this scenario and others described in our [motivating examples on the website.](https://rt-sketch-anon.github.io/additional.html#why_sketches)

---

> > ### Comment · Reviewer_GDgQ · 2023-11-19
> >
> > The reviewer would first like to let the authors know that their hard work is appreciated. However, my rating has not changed.
> >
> >
> > This rebuttal was hard to read as it was not organized well with much of the text being repeated multiple times. For example, the metrics are discussed in the second post, then a point about sketch ambiguity is repeated despite it already being discussed in a previous post, and then the metrics discussion is repeated again in the last post.
> >
> > ---
> >
> > # On Sketches as a Goal
> >
> > I appreciate the different perspectives between the authors and the other reviewers regarding the utility of sketches as a goal specification. I do think sketches can be useful for goal specifications. However, in the form that the authors presented in this work, sketches seem more laborious, and there doesn't seem to be a big enough boost in the results when compared to using language as a goal.
> >
> > As mentioned previously, for the motivating problems it would still seem like the desirable solution would be to create more intelligent agents to create reasonable solutions or ask for clarifications when given an ambiguous problem.
> >
> > ---
> >
> > # Inaccuracy in the Shared Response
> >
> > > " We want to reiterate that this is the first work to our knowledge to consider sketches as a modality for goal specification, let alone any representation to this degree of visual abstraction."
> >
> > This may have been an inaccuracy as the authors may have meant this statement only for robotic manipulation tasks. In the manuscript, the authors write
> > "Prior work [1] has shown the utility of sketches over pure language commands in navigation scenarios. "
> >
> > However, using sketches as a goal specification has also been investigated before for robot manipulation [2,3].
> >
> >
> > # Metrics and Results Concern
> >
> > The new visualizations are appreciated and they help clarify the metrics. However, concerns regarding the metrics and the results still remain.
> >
> > (1) The differences in results in the spatial precision are very close, a difference of a few pixels. Errors from human annotators in choosing the best frame and the centroid could lead to pixel-sized errors. In combination, these concerns make it hard to determine if any method performs best in Table 1's Spatial Precision results.
> >
> > (2) Choosing distance in the pixel space seems like it could also add errors to this study. For example, a displacement of 10cm from the back of the table will have a different pixel displacement than at the front of the table.
> >
> > (3) For the spatial precision metric, it would seem that the choice of having a human choose the most aligned image rather than where the robot finished goes against the motivation of this work. If I gave a sketch of a table arrangement and then at some point, the robot got this correct but didn't stop and ruined the arrangement, I would consider it to have failed.
> >
> > (4) Does "excessive retrying" also occur if the robot keeps aligning the scenes, tries too often, but does not undo its progress?
> >
> > ---
> >
> > [1] Porfirio, David, et al. "Sketching Robot Programs On the Fly." Proceedings of the 2023 ACM/IEEE International Conference on Human-Robot Interaction. 2023.
> >
> > [2] Cui, Yuchen, et al. "Can Foundation Models Perform Zero-Shot Task Specification For Robot Manipulation?." Learning for Dynamics and Control Conference. PMLR, 2022.
> >
> > [3] Barber, Christine M., et al. "Sketch-based robot programming." 2010 25th International Conference of Image and Vision Computing New Zealand. IEEE, 2010.

---

> ### Author Response · Authors · 2023-11-18
> **Response to Reviewer GDgQ (cont.)**
>
> > While language can be ambiguous, so can sketches. As an example, if a sketch was given with an empty table, am I telling the agent that I want it to throw out the garbage on the table or am I telling it to ignore the garbage? It would seem like the desirable solution would be to create more intelligent agents to create reasonable solutions or ask for clarifications when given an ambiguous problem.
>
> Absolutely, we fully agree that sketches have their own drawbacks and limitations. Specifically, sketches with imperfect or omitted details can be highly ambiguous or difficult to interpret. Earlier, we discuss a [sketch-and-language conditioned policy](https://rt-sketch-anon.github.io/additional.html#multimodal) we have since implemented which can address limitations of either modality alone. In your example, language might help disambiguate a sketch by distinguishing between ignoring or throwing away objects, but a sketch could provide more spatial precision than language alone for placing a new object on the empty table, as we demonstrate [here](https://rt-sketch-anon.github.io/additional.html#multimodal).
>
> Another promising future direction is treating sketches in a more general sense (one could imagine drawing an ‘X’ over irrelevant objects or those that should be discarded as you suggested, using arrows to indicate where objects should be placed, or using sketches as general-purpose constraints or cost-functions, for instance). For all of these future directions, showing the merits of sketches in the first place is a prerequisite.

---

> ### Author Response · Authors · 2023-11-18
> **Response to Reviewer GDgQ (cont.)**
>
> > "Spatial Precision" and "Failure Occurrence" are not carefully defined or motivated. Re: object centroids, why manual keypoint annotation is necessary over using an off-the-shelf image classifier should have been mentioned.
> Thanks for bringing these up! We apologize for the lack of clarity and while we respond to this in the shared response above, we provide a more thorough explanation here:
>
> Spatial precision denotes the difference in the achieved versus desired placement of an object in terms of pixel-wise distance. There are two challenges: (1) determining when the robot achieves alignment with the goal (which may not be the last frame due to retrying attempts that disturb the scene), and (2) measuring the target object’s achieved vs. desired position. Choosing the most aligned frame (1) is a notoriously difficult video temporal segmentation problem, and while an out-of-the-box object detector is an option for (2), we want to avoid conflating object detection errors with policy imprecision.
>
> From an implementation standpoint, open-vocabulary detectors may not handle all language instructions robustly (especially for our ambiguous language experiments), detection thresholds may be brittle and difficult to tune across rollouts, and the visual scenes contain robot arm occlusions and distractors which pose significant challenges. Thus, we have a human annotator manually annotate the most aligned frame, along with 2D keypoints for the salient object centroid in this frame and in the ground truth goal image, and measure RMSE. **We have since added visualizations of this metric to the Appendix and website**.
>
> Regarding failure occurrence, we are specifically referring to excessive re-trying — in which the robot aligns the scene but then retries too often and undoes its progress ([video](https://rt-sketch-anon.github.io/additional.html#failure)). We apologize for the unclear wording in Table 1 – we have since updated the wording and note that the reported numbers are the proportion of rollouts in which this failure mode occurs. We report this somewhat surprising failure mode due to the frequency with which the goal-image policy exhibits this behavior, as a result of trying to over-align a scene in pixel space. Meanwhile, the sketch-conditioned policy is far less sensitive, likely having learned a much higher-level notion of alignment due to the abstraction of sketches.
>
> ***
> > It is not obvious how big of a difference the errors are when looking at the RMSE in pixels. Could the errors when finding the centroid and the manual keypoint annotations be an issue in measuring this? Can a visualization be created to show this?
>
> We agree that the numbers alone are difficult to contextualize without a visualization, and have since updated our Appendix and website [here](https://rt-sketch-anon.github.io/additional.html#eval_metrics) to illustrate the discrepancy in achieved vs. target object placements in different images, with which we measure RMSE. As in our response above, we think that using manual keypoint annotation actually allows us to more accurately quantify policy imprecision without confounding with object detection errors.
> ***
> > Typically bolded numbers in a column/row represent the method with best performance. The meaning of the shading of the cells of Table 1 adds a lot of confusion.
>
> Thanks for your attention to detail and apologies for the confusion. We have since changed the bolding/shading in order to obey standard conventions and updated the caption.

---

> ### Author Response · Authors · 2023-11-21
> **Follow-Up to Reviewer GDgQ (1/3)**
>
> > This rebuttal was hard to read as it was not organized well with much of the text being repeated multiple times.
>
> Really sorry for the inconvenience; we’ve since re-organized our rebuttal. A section of our response accidentally got duplicated in trying to fit our response within the character count limit – hopefully it is more clear now.
> ***
> ### On Sketches as a Goal
> > In the form that the authors presented in this work, sketches seem more laborious, and there doesn't seem to be a big enough boost in the results when compared to using language as a goal.
>
> Figure 3, H4 demonstrates that sketches improve perceived spatial/semantic alignment by 2.8x and 2.4x, respectively in the settings where language is ambiguous. In addition, producing the line sketches used in evaluation takes only a few seconds since a sketch can have as little as a few salient object outlines in order for the policy to effectively interpret the sketch and perform the tasks (H2 experiments).
> ***
>
> > As mentioned previously, for the motivating problems it would still seem like the desirable solution would be to create more intelligent agents to create reasonable solutions or ask for clarifications when given an ambiguous problem.
>
> We agree that having a truly interactive mode of goal-specification down the line, with multimodal input and closed-loop feedback, is extremely compelling! However, we argue that realizing a multimodal, interactive agent requires first demonstrating the merits of individual modalities, and sketches are critically underexplored in robotic manipulation. The focus of this work is to understand the benefits and challenges of using sketches as an input modality.
> ***
> ### Inaccuracy in the Shared Response
> > "We want to reiterate that this is the first work to our knowledge to consider sketches as a modality for goal specification, let alone any representation to this degree of visual abstraction."
> > This may have been an inaccuracy as the authors may have meant this statement only for robotic manipulation tasks. In the manuscript, the authors write "Prior work [1] has shown the utility of sketches over pure language commands in navigation scenarios. "
> > However, using sketches as a goal specification has also been investigated before for robot manipulation [1,2,3].
>
> This was intended in the context of manipulation.  We thank you for pointing out [1,2,3], which are in fact relevant. We have since updated the related work and contributions in the [draft](https://rt-sketch-anon.github.io/assets/rt_sketch_revised.pdf) with these in mind, with changes denoted in red. Below, we point out several key differences that distinguish our work:
>
> [2] does not train the policy to take sketch images as input, and came to the conclusion that the scene image is better than the sketch image at goal specification. They note: “We also observe that using same scene images leads to better results than using internet images or drawings, which is along expected lines due to smaller domain gap.” Our result is different and complementary, in that policies trained to take sketches as input outperform a scene image conditioned policy, by 1.63x and 1.5x in terms of Likert ratings for perceived spatial and semantic alignment, subject to visual distractors (H3 experiments).
>
> [1] and [3] leverage sketching interfaces in a very different way from us, using sketches primarily to indicate paths for navigation. [1] additionally uses sketches for specifying path-planning constraints, but uses significant heuristic processing by detecting the outline of a sketch, fitting an oriented bounding box, and de-projecting this to the 3D workspace to indicate collision zones to a planner. We in contrast use a raw sketch. Additionally, their “sketching” interface for manipulation involves loading a visualization of the robot model, and using lines to indicate how the joints should each move. The authors report that a common user complaint was the interface “not feeling natural.” These kinds of interfaces place an inconvenient burden on the user to provide low-level guidance for planning and manipulation via sketches. [1] specifically suggests that an avenue for future work is a more abstract design where “users sketch the semantic of an action, rather than the motion of its components. An example would be to circle the object to be picked up and to sketch a line from the gripper to the object. An inbuilt algorithm would then compute the optimal arm motion.” In this work, we move towards these far more abstract representations in the form of scene sketches.
>
> These observations are just to say that there is ample room for exploration and improvement around using sketches, and our work takes a novel perspective of using simplified sketches of the goal specification in a learning-based manipulation policy.
> ***

---

> ### Author Response · Authors · 2023-11-21
> **Follow-Up to Reviewer GDgQ (2/3)**
>
> ### Metrics and Results Concern
> > (1) The differences in results in the spatial precision are very close, a difference of a few pixels. Errors from human annotators in choosing the best frame and the centroid could lead to pixel-sized errors. In combination, these concerns make it hard to determine if any method performs best in Table 1's Spatial Precision results.
>
> The average range of errors we see across policies under the RMSE metric is ~2-8, looking at Table 1. This  may seem small, but actually captures significant disparity. A value on the high side like 8, which is the typical imprecision we see for RT-1 and RT-Goal Image in H3/H4 settings, corresponds to around ~8**2 so ~64px L2 distance. For the small image size we consider (320 x 256), this is being off by about 20% (64/320) of the image dimensions, which is quite large. On the website, we show [an example of a failed rollout with an RMSE of 10](https://rt-sketch-anon.github.io/additional.html#eval_metrics), which is really off by ~31% of the image dimensions and is a substantial difference! The green chip bag is fully opposite where it should be. Thus, RT-Sketch’s average spatial precision in the range of 3.02 to 5.36 suggests that these severe failures are uncharacteristic. On average, our policy is off by just ~3-10% of image dimensions, compared to errors on the order of ~20+% of image dimensions for baselines in H3/4.
>
> We acknowledge that some amount of human labeling error can occur when annotating keypoints manually, but a human would have to be off by a lot to make mistakes at the severity of 20-30% of image dimensions.
>
> ***
>
> > (2) Choosing distance in the pixel space seems like it could also add errors to this study. For example, a displacement of 10cm from the back of the table will have a different pixel displacement than at the front of the table.
>
> To ensure that using the metric to compare policies is not biased by perspective, we use the same initial and final scene configurations to evaluate all policies. Therefore, no policy is biased by objects being placed at the back vs. front of the table.
> ***
> > (3) For the spatial precision metric, it would seem that the choice of having a human choose the most aligned image rather than where the robot finished goes against the motivation of this work. If I gave a sketch of a table arrangement and then at some point, the robot got this correct but didn't stop and ruined the arrangement, I would consider it to have failed.
> (4) Does "excessive retrying" also occur if the robot keeps aligning the scenes, tries too often, but does not undo its progress?
>
> If the robot achieved alignment but then didn’t stop and ruined the arrangement, this is considered excessive retrying. We would still compute RMSE over the frame when alignment is achieved, but the trial is considered a failure due to excessive retrying (we report both task success rate and RMSE).
>
> Your question insightfully hints at two different but important aspects of understanding policy performance:
> * 1. does the policy align the scene?
> * 2. when the policy aligns the scene, to what degree is it off?
>
> Here, we can see that task success rate gets at (1) but doesn’t say anything about precision (2), while RMSE gets at (2) but doesn’t guarantee terminal scene alignment (1). We argue that both are necessary to understand the performance of any goal-conditioned policy. We are excited that RT-Sketch’s decent performance under both metrics indicates the promise of sketches.
>
> ***
> ### Remarks
> We again apologize if the format of rebuttal was not clear, and our statement about the use of sketches was not specific enough to emphasize our contribution of developing a learning-based sketch-conditioned policy that enables convenience and alternative abstraction of goals.
>
> Regarding evaluation metrics, we do believe our metrics are going beyond a large number of prior works in goal-conditioned manipulation that often only report task success. Our Likert, spatial precision, and failure metrics attempt to provide a more detailed representation of the type of successes achieved using any of the goal specification modalities (language, goal images, sketches) both semantically and spatially. What the reviewer has proposed in terms of object-detection might conflate performance with detection error.
>
> We respect your decision ultimately, but ask that you kindly factor in our thorough attempt to address your concerns. We are excited about the potential that sketches can provide as a modality for goal specification, and about the findings of our paper demonstrating these capabilities in manipulation, and hope future work can use sketches in tandem with other modalities. We’d greatly appreciate constructive suggestions on what would help change your opinion about the paper and score.

---

> > ### Author Response · Authors · 2023-11-21
> > **Follow-Up to Reviewer GDgQ (3/3)**
> >
> > ### References
> > [1] Porfirio, David, et al. "Sketching Robot Programs On the Fly." Proceedings of the 2023 ACM/IEEE International Conference on Human-Robot Interaction. 2023.
> >
> > [2] Cui, Yuchen, et al. "Can Foundation Models Perform Zero-Shot Task Specification For Robot Manipulation?." Learning for Dynamics and Control Conference. PMLR, 2022.
> >
> > [3] Barber, Christine M., et al. "Sketch-based robot programming." 2010 25th International Conference of Image and Vision Computing New Zealand. IEEE, 2010.
> >
> > [4] Zeng, Andy, et al. "Socratic models: Composing zero-shot multimodal reasoning with language." arXiv preprint arXiv:2204.00598 (2022).

---

### Official Review · Reviewer_fq96 · 2023-10-31

**Soundness:** 3 good
**Presentation:** 3 good
**Contribution:** 3 good
**Rating:** 6
**Confidence:** 4

**Summary:**

The authors present RT-Sketch, a goal-conditioned policy that takes a hand-drawn sketch of the desired scene as input and outputs actions. They train RT-Sketch on a dataset of paired trajectories and corresponding synthetically generated goal sketches. The experimental results show that RT-Sketch performs on a similar level to image or language-conditioned agents in straightforward settings, while achieving greater robustness when language goals are ambiguous or visual distractors are present.

**Strengths:**

- This study introduces a novel method that employs sketches as the target for conditioned imitation learning. This approach is well-motivated as sketches can be more advantageous than language and natural images in situations involving language ambiguity and visual distractors.

- The proposed method was validated through experimental settings that aimed to address four hypotheses. These hypotheses were concerned with whether sketches are expressive enough (H1), if the proposed method can handle various abstraction levels of sketches (H2), whether sketches are robust enough to tackle distractors compared to goal images (H3), and if sketches outperform ambiguous language (H4). The results of these experiments were convincing to me.

- The paper has been skillfully crafted and is presented in a manner that is both clear and concise.

**Weaknesses:**

To obtain the sketch that can precisely convey the goal is essential to the success of the proposed method, I have the following concerns which may limit the practical usage of the proposed method:

1. Given the potential (vast) cost associated with collecting human sketches for various scenarios, perhaps scalability could be a concern when applying this approach to broader scenarios, rather than solely relying on the benchmark presented in this study.

2. The authors did not conduct experiments to determine how various image-to-sketch generation methods can affect the final results of RT-Sketch, which could potentially create a bottleneck in the entire pipeline.

**Questions:**

Please see the weaknesses.

---

> ### Author Response · Authors · 2023-11-18
> **Response to Reviewer fq96**
>
> We are glad to hear that you are encouraged by the novelty of sketches and find our experiments compelling! Thanks for your comments.
> ***
>
> > Given the potential (vast) cost associated with collecting human sketches for various scenarios, perhaps scalability could be a concern when applying this approach to broader scenarios, rather than solely relying on the benchmark presented in this study.
>
> Scalability is an important consideration, and we kindly ask that you refer to our shared response above for a more thorough response. To highlight a few things, we train our image-to-sketch GAN on 500 examples, which is a mere 0.6% of the overall demonstration trajectory dataset size. This amount of examples is also on par with datasets in vision literature; namely the original PhotoSketching GAN [1] trains on 5,000 sketches, and ControlNet [2] is trained with 1K examples at the minimum. While 500 may still seem like a lot, in reality it took a few hours of data collection over 3 days, and we believe the benefits of training this GAN are worth it if the end result is a much more convenient means of goal specification which  takes seconds to provide.
>
> Also, while goal-image datasets are very much tied to the environment they were collected in, we could certainly train our image-to-sketch network on the same amount of images spanning different datasets. The original PhotoSketching GAN which we fine-tuned was actually trained on internet-scale data with varied categories of people, animals, etc., so training this network with multiple robot environments is certainly within scope. This would be one straightforward way to scale sketch-based IL to multiple robot setups without the need to collect any additional demonstration or sketch data.
>
> ***
> > The authors did not conduct experiments to determine how various image-to-sketch generation methods can affect the final results of RT-Sketch, which could potentially create a bottleneck in the entire pipeline.
>
> As you point out, the choice of what technique we use for image-to-sketch generation is critical to the overall success of the pipeline. We did in fact experiment with various approaches for image-to-sketch generation before converging on the proposed method. We have since updated our website with [additional results for alternate image-to-sketch approaches](https://rt-sketch-anon.github.io/additional.html#alt_methods) we tried, and their pitfalls. A critical challenge is that we want generated sketches to be minimal in nature (i.e. background objects are ignored, rough outlines are acceptable). Within the vision and graphics communities, the most recent work on image-to-sketch generation (CLIPasso, CLIPAscene) is concerned with producing aesthetically pleasing results, which is not a priority in our setting. We find that a GAN-based approach gives us control over the kinds of output sketches we would like, and allows us to make use of existing large-scale, crowdsourced data of paired images and sketches while just fine-tuning on our own data.
>
> We also acknowledge that sketch-to-image generation is another possibility to consider. We did consider state-of-the-art models like ControlNet and other Diffusion-type variants for image synthesis, but the challenge here is still overcoming hallucinations in generated images and jointly enabling style transfer (generated images must match the style of robot images if we want to condition policies on them). That said, generative models for image synthesis are ever-improving, and the RT-Sketch policy architecture is agnostic to the exact choice of visual input. In the future, we are excited about revisiting these techniques for sketch-based IL, but we hope that this provides more context into our design choices for the time being.
>
> ***
>
> [1] Li, Mengtian, et al. "Photo-sketching: Inferring contour drawings from images." 2019 IEEE Winter Conference on Applications of Computer Vision (WACV). IEEE, 2019.
>
> [2] Zhang, Lvmin, Anyi Rao, and Maneesh Agrawala. "Adding conditional control to text-to-image diffusion models." Proceedings of the IEEE/CVF International Conference on Computer Vision. 2023.

---

> > ### Author Response · Authors · 2023-11-21
> > **Follow-Up to Reviewer fq96**
> >
> > As the rebuttal period comes to a close, we hope we have addressed your concerns. We would greatly appreciate it if you could consider raising your score in light of the response provided above and our [additional results](https://rt-sketch-anon.github.io/additional.html) since submission. Thanks for your feedback!

---

### Official Review · Reviewer_1JK3 · 2023-10-31

**Soundness:** 3 good
**Presentation:** 3 good
**Contribution:** 3 good
**Rating:** 8
**Confidence:** 4

**Summary:**

This study introduces RT-Sketch, a novel approach to visual imitation learning that utilizes hand-drawn sketches for goal specification. Unlike ambiguous natural language or overly detailed images, sketches strike a balance by being user-friendly and spatially aware. RT-Sketch achieves performance similar to image or language-conditioned agents in straightforward scenarios but excels in handling ambiguity and visual distractions. It demonstrates the capacity to interpret and act upon sketches of varying specificity, highlighting their versatility in goal representation.

**Strengths:**

+ Overall, this work opens a new direction for leveraging sketches for goal-conditioned imitation learning.
+ The motivation is clear and reasonable.
+ Use of Contour Drawing Dataset is logical and helps to mitigate the gap between synthetic sketches and real freehand sketches.

**Weaknesses:**

## Missing subsection in the related work:
In recent years, there has been a significant body of work at the intersection of sketches for visual understanding. I would suggest the author add one separate subsection discussing a few major works on 'Sketch for Visual Understanding' in the related work parts. The authors could use this for their reference: https://github.com/MarkMoHR/Awesome-Sketch-Based-Applications/blob/master/README.md. Some representative works include:
- a) https://arxiv.org/abs/2303.15149, CVPR'23
- b) https://arxiv.org/pdf/2302.05543.pdf (ControlNet uses sketch for image generation), ICCV'23
- c) https://arxiv.org/pdf/2303.11502.pdf, CVPR'23
- d) https://arxiv.org/abs/2203.14843, CVPR'22.
- e) https://arxiv.org/abs/2204.11964, CVPR'23

## Freehand sketches vs Synthetic Sketches/Edgemaps
Free-hand sketches and edge maps are different, and many existing works on sketches have claimed that models trained from edge maps do not generalize well to free-hand sketches. Some relevant works are https://openaccess.thecvf.com/content/CVPR2023/papers/Koley_Picture_That_Sketch_Photorealistic_Image_Generation_From_Abstract_Sketches_CVPR_2023_paper.pdf and https://github.com/mtli/PhotoSketch. Some discussions around that could be helpful.

## Minor
- Some self-contained caption could be helpful for Fig. 2.

**Questions:**

1. The experiment section could be made a little more self-contained so that it would be easier to digest for readers from a broader background. I wonder if the authors could pay some attention to that.
2. Figure 3 could benefit from a more comprehensive caption to guide the reader through the observations, making it easier to understand.
3. Is it possible to add more visual examples where a sketch is found to be better than a text-only counterpart?

---

> ### Author Response · Authors · 2023-11-18
> **Response to Reviewer 1JK3**
>
> Thank you for your appreciation of sketches and our technical approach!
> ***
> > Missing subsection in the related work
>
> Thanks for putting these alternate sketch-image translation approaches on our radar. In the [updated draft](https://rt-sketch-anon.github.io/assets/rt_sketch_revised.pdf), we have since referenced them throughout Related Works and greatly appreciate the references.
> ***
> > Free-hand sketches and edge maps are different, and many existing works on sketches have claimed that models trained from edge maps do not generalize well to free-hand sketches. Some discussions around that could be helpful.
>
> This is a great point. We do fully acknowledge that there is a potentially large domain gap between edge-detected / GAN-generated images and manually drawn sketches, but in the absence of manual sketches at scale, we make certain design choices to make use of modalities available to us. Specifically, we train RT-Sketch on a mix of GAN-generated sketches (60%) to be able to handle hand-drawn sketches at test-time, but we include edge-detected images (20%) and goal images (20%) to encourage the policy to afford variations in style, shading, and level of detail in sketches.
>
> Empirically, we do find that RT-Sketch performs well on hand-drawn sketches at test-time, as validated by H1-4, despite being trained on these non hand-drawn representations. This finding suggests that our image-to-sketch GAN produces reasonably realistic sketches in the first place, such that real sketches do not throw the policy off at test-time.
>
> We are also encouraged, somewhat surprisingly, that the policy can handle colored sketches and completely free-hand drawings, despite not being trained on such representations. Since training on GAN-generated line sketches alone would be a huge distribution shift from these kinds of sketches, this suggests that our affine/color augmentations and the inclusion of other  modalities (edges/goal images) enables this emergent robustness. Prior work in sketch-based object detection has found that similar kinds of augmentations can help bridge the domain gap [1].
>
> Finally, we note that we use a Transformer backbone [2], and Transformers have been shown to have a high model capacity; we posit that this might enable the policy to interpret these different modalities well.
> ***
> > Some self-contained caption could be helpful for Fig. 2.
>
> Thank you! We agree and have updated the caption.
> ***
> > The experiment section could be made a little more self-contained so that it would be easier to digest for readers from a broader background. I wonder if the authors could pay some attention to that.
> Thanks for your suggestion! For the time being we have added some clarifications to the experiments section regarding our evaluation procedure and metrics, but we can certainly make this section more concise in a camera-ready version if accepted.
> > Figure 3 could benefit from a more comprehensive caption to guide the reader through the observations, making it easier to understand.
>
> We have since updated the caption for Figure 3, with changes highlighted in blue, but please let us know if anything is still unclear.
> ***
> > Is it possible to add more visual examples where a sketch is found to be better than a text-only counterpart?
>
> Absolutely, we agree that the motivation for sketches could be made more clear. We have since addressed this in the shared response as well as in this [section on the website](https://rt-sketch-anon.github.io/additional.html#why_sketches) where we illustrate scenarios in which sketches could either complement or address limitations of language alone.
>
> ***
> [1] Chowdhury, Pinaki Nath, et al. "What Can Human Sketches Do for Object Detection?." CVPR 2023.
>
> [2] Vaswani, Ashish, et al. "Attention is all you need." NeurIPS 2017.

---

### Author Response · Authors · 2023-11-18
**Shared Response to All Reviewers**

Thanks to all the reviewers for providing such thoughtful feedback! We are glad to hear that the reviewers appreciate the novelty of sketches as a modality for goal specification, our thorough experimentation, and the clarity of the paper. We first address shared key concerns:
***
### **Role of Sketches**
Reviewers GDgQ and rbwp seek to better understand the role of sketches as a modality for goal specification. We argue that sketches have a number of benefits: 1) they provide spatial awareness, 2) they allow for robustness to visual distractors, and 3) they are convenient and intuitive to provide on-the-fly. We want to reiterate that this is the first work to our knowledge to consider sketches as a modality for goal specification, let alone any representation to this degree of visual abstraction. We view sketches as a step towards a far more flexible goal specification beyond (or in addition to) language and goal images, without compromising on performance. To better contextualize the value of sketches, we have **since added** [motivating examples](https://rt-sketch-anon.github.io/additional.html#why_sketches) for sketches to our website. That said, we do not view sketches as a standalone or optimal interface. We agree with reviewers about the promise of multimodal goal specification, and have since added **new experiments with a** [sketch-and-language conditioned manipulation policy](https://rt-sketch-anon.github.io/additional.html#multimodal) which achieves better semantic alignment over a sketch-alone policy, and better spatial alignment over a language-alone policy.
***
### **Evaluation**
We apologize if details about our evaluation procedure and metrics were not clear during submission, and have **since updated the paper and Appendix with metrics clearly defined, motivated, and visualized**. A few critical clarifications:
* While RT-Sketch is trained on synthetically generated GAN sketches, **we use real, hand-drawn sketches (drawn by a single annotator) as the input across all rollouts**
* We evaluate RT-Sketch with a Likert-based human assessment of 62 individuals, split so that 8-12 people evaluate each of the 6 skills.

Regarding metrics, the notion of goal alignment does not have standardized metrics in prior work, leading us to use the following metrics:
* ***Spatial precision*** refers to the degree to which a policy achieves alignment with a visual goal. Here, we measure the centroid of a manipulated object in an achieved vs. goal image, since the rest of the scene is usually quite static. We use a manually annotated keypoint for the object centroid (as opposed to an output from an out-of-the-box object detector) specifically to avoid conflating errors in *detection* with *policy imprecision.*
* ***Failure occurrence*** specifically refers to the tendency of (mostly goal-image conditioned) policies to exhibit [excessive retrying behavior](https://rt-sketch-anon.github.io/additional.html#failure). We find that sketch-based policies are far less susceptible to this, as they do not reason at this unnecessary level of granularity.
We hope that reviewers appreciate our efforts to evaluate something as subjective as sketches as fairly and rigorously as possible.
***
### **Generalization**
To [stress-test generalization](https://rt-sketch-anon.github.io/additional.html#generalization) to sketches drawn by different people, we have **since evaluated RT-Sketch with 22 evaluators on 30 sketches drawn by 6 different people**. RT-Sketch achieves high perceived spatial alignment, on par with the original sketches we evaluated on (drawn by only one person), and with no significant dropoff between people.
***
### **Scalability**
While we train an image-to-sketch translation network on 500 manually collected sketches, this is a modest **0.6%** of the total number of demonstration trajectories, and only took a few hours spread over 3 days. This small upfront cost ultimately enables an extremely convenient way of doing downstream goal specification. We’d like to emphasize that  nothing in our approach is tied to a particular robot or dataset; we propose a general framework for training a sketch-to-action agent that is agnostic to both the choice of robot and choice of underlying IL algorithm. In principle, we could train our image-to-sketch translation network on multiple robot embodiment datasets, and readily apply our framework to existing manipulation datasets to enable sketch-based IL.

---

> ### Author Response · Authors · 2023-11-18
> **Summary of Updates**
>
> In short, our main updates are:
>
> **Additional results**
> * New results with a sketch-and-language-conditioned policy which empirically shows better performance than either modality alone and illustrates the compatibility of sketches with other modalities
> * New experiments with 30 crowdsourced sketches from 6 non-expert and untrained individuals to test RT-Sketch generalization
>
> **[Supplementary material](https://rt-sketch-anon.github.io/additional.html) on the website**
> * Videos of the above new results
> * Motivating examples for considering sketches
>
> **An [updated draft](https://rt-sketch-anon.github.io/assets/rt_sketch_revised.pdf) indicating all new writing changes in blue**
> * A more detailed discussion of the role of sketches
> * Explanation on evaluation protocol
> Visualizations of evaluation metrics
> Visualization of alternate image-to-sketch generation methods

---

### Meta-Review · Area_Chair_4yPe · 2023-12-11

**Metareview:**

This paper was reviewed by three experts and received mixed scores. Though all reviewers agree some aspects of the paper are promising, they also consistently raise concerns listed below.

1. The motivation for using sketches to specify a goal is unclear.

2. The experiments are limited. More real-world experiments are required to demonstrate the practice of this approach.

While the research demonstrated indeed has promise, the decision is not to recommend acceptance in its current state. The authors are encouraged to consider the reviewers' comments when revising the paper for submission elsewhere.

**Justification For Why Not Higher Score:**

1. The motivation for using sketches to specify a goal is unclear.

2. The experiments are limited. More real-world experiments are required to demonstrate the practice of this approach.

**Justification For Why Not Lower Score:**

NA

---

### Decision · Program_Chairs · 2024-01-16

Reject